# Spatial-linked alignment tool (SLAT) for aligning heterogenous slices

Chen-Rui Xia[1,2,4], Zhi-Jie Cao ●[1,2,4] ✉, Xin-Ming Tu[1,3] & Ge Gao ●[1,2] ✉

Spatially resolved omics technologies reveal the spatial organization of cells in various biological systems. Here we propose SLAT (Spatially-Linked Alignment Tool), a graph-based algorithm for efficient and effective alignment of spatial slices. Adopting a graph adversarial matching strategy, SLAT is the first algorithm capable of aligning heterogenous spatial data across distinct technologies and modalities. Systematic benchmarks demonstrate SLAT's superior precision, robustness, and speed over existing state-of-the-arts. Applications to multiple real-world datasets further show SLAT's utility in enhancing cell-typing resolution, integrating multiple modalities for regulatory inference, and mapping fine-scale spatial-temporal changes during development. The full SLAT package is available at https://github.com/gao-lab/SLAT.

Recently emerging spatial omics technologies enable profiling the location, intercommunication, and functional cooperation of native cells through fluorescence in situ hybridization (seqFISH[1], MERFISH[2], seqFISH+[3] and Xenium[4]) and spatial barcoding (10× Visium[5], HDST[6], Slide-seqV2[7], Stereo-seq[8] and spatial-ATAC-seq[9]) from multiple tissue "slices," revealing tissue structure heterogeneity and shedding light on the underlying physiological and pathological mechanisms[8,10].

Properly aligning cells that share common molecular identity (e.g., cell type) and spatial context across multiple slices, especially these generated from distinct sources, is critical for their follow-up analysis. For example, inter-technology alignment effectively bridges technologies complementary in spatial resolution and omics coverage[4], while aligning various time points during spatially dynamic processes like embryogenesis helps identify key spatial temporal changes and their molecular underpinnings. However, current spatial alignment algorithms[11–13] are mostly designed for homogeneous alignments (e.g., three-dimensional reconstruction from consecutive slices[14,15]), and can hardly handle heterogeneous slices which often involve complex non-rigid deformations, uneven spatial resolutions as well as complex batch effects.

Here, we introduce SLAT (Spatially-Linked Alignment Tool), a unified framework for aligning both homogenous and heterogeneous single-cell spatial datasets. By modeling the intercellular relationship as a spatial graph, SLAT adopts graph neural networks and adversarial matching for robustly aligning spatial slices. In addition to its superior performance revealed by systematic benchmarks, as the first algorithm capable of aligning heterogenous spatial data, SLAT introduces a wide range of application scenarios including alignment across distinct technologies and experimental conditions. SLAT is publicly accessible at https://github.com/gao-lab/SLAT and will be continuously updated as spatial omics technologies evolve.

## Results

### Align heterogenous spatial omics data via graph adversarial matching

By modeling the spatial topology per slice as a spatial graph where each cell is connected to its nearest neighbors by edges, we reformulate the slice-alignment task as a graph-matching problem. In efforts to correct potential cross-dataset batch effect, SLAT employed a Singular Value Decomposition (SVD)-based strategy to project omics profiles of the cells into a shared low-dimensional space ("Methods"), which in turn serves as node features of the spatial graphs.

Multilayer lightweight graph convolutional networks are incorporated to propagate and aggregate information between cells and their neighbors via stepwise concatenations, generating a holistic representation with information at multiple scales from individual cells to local niches as well as global positions. Then, SLAT solves a minimum-cost bipartite matching problem between the spatial graphs

[1]State Key Laboratory of Protein and Plant Gene Research, School of Life Sciences, Biomedical Pioneering Innovative Center (BIOPIC) and Beijing Advanced Innovation Center for Genomics (ICG), Center for Bioinformatics (CBI), Peking University, 100871 Beijing, China. [2]Changping Laboratory, 102206 Beijing, China. [3]Present address: Paul Allen School of Computer Science and Engineering, University of Washington, Seattle, WA 98195, USA. [4]These authors contributed equally: Chen-Rui Xia, Zhi-Jie Cao. ✉e-mail: caozj@mail.cbi.pku.edu.cn; gaog@mail.cbi.pku.edu.cn

through a dedicated adversarial component[16] to align cells from different slices (Fig. 1, "Methods").

Notably, apart from congruent regions that can be well aligned across slices, heterogeneous alignment often involves distinct regions that reflect biologically relevant spatial alterations. To avoid artificially aligning such regions (i.e., over-alignment), SLAT adopts an adaptive clipping strategy during adversarial matching to retain only the closest pairs between slices in terms of their cosine similarity in the embedding space, taken as reliable anchors for guiding the whole alignment procedure (Supplementary Fig. 1). In particular, SLAT utilizes an empirical probabilistic matching strategy and returns matches with high confidence only (default $p$ value threshold = 0.05, see "Methods").

## Systematic benchmarks suggest SLAT is accurate robust and fast

To evaluate SLAT's performance over existing algorithms which are designed for homogeneous alignment[12,14], systematic benchmarks were conducted based on consecutive, homogeneous slices from the same tissue generated by three representative technologies spanning a wide range of throughput, resolution, and technological routes: 10× Visium[5], MERFISH[2], and Stereo-seq[8] (Fig. 2a, Supplementary Table 1, and Supplementary Fig. 2).

We first ran synthetic tests where a spatial slice and its rotated and noise-perturbed copy were fed into the alignment algorithms ("Methods"). We examined how the alignments could be used to correct the artificial rotation as well as how they compare with the known ground truth matching (Supplementary Fig. 3). Both SLAT and PASTE achieved high accuracy, while the performance of spatially unaware algorithms Seurat and Harmony degraded substantially with increasing levels of noise (Supplementary Fig. 3). To our surprise, STAGATE produced low accuracy across all noise levels, which could be attributed to its lack of noise reduction and batch correction components.

We then tried to align pairs of distinct slices in the same datasets (Supplementary Fig. 4). Due to the lack of ground truth matchings, we quantify alignment accuracy by the fraction of cells correctly matched in expert-curated cell types and spatial regions, respectively (Fig. 2b, c, also see Supplementary Fig. 5), and the joint accuracy is defined as the fraction of cells where both cell types and spatial regions are correctly matched (Fig. 2d).

Overall, SLAT outperformed PASTE[14] and STAGATE[12] in all three datasets (Fig. 2b–d). Of note, PASTE exhibited particularly low cell type accuracy in the MERFISH dataset (Fig. 2c and Supplementary Fig. 5) where different cell types are spatially interlaced (Supplementary Fig. 6), probably due to its excessive reliance on the spatial distance between cells over molecular features. Consistent with the synthetic test, we found that STAGATE produced suboptimal alignments, while its performance improved for split slices which are free of batch effects ("Methods," Supplementary Fig. 7). By combining spatial context with transcriptome profiles, SLAT is better equipped to distinguish transcriptionally similar but spatially distinct cell groups. Consistently, we found that SLAT also achieved significantly higher alignment accuracy than the conventional spatially unaware algorithms Seurat[17] and Harmony[18] (Fig. 2c, d). In particular, although these algorithms matched cell types reasonably well, their alignment largely disarranged spatial regions (Fig. 2b and Supplementary Figs. 4 and 5). The comparisons were also consistent in alternative metrics based on micro- and macro-F1 scores (Supplementary Fig. 8), as well as the correction of artificial rotation (Supplementary Fig. 9).

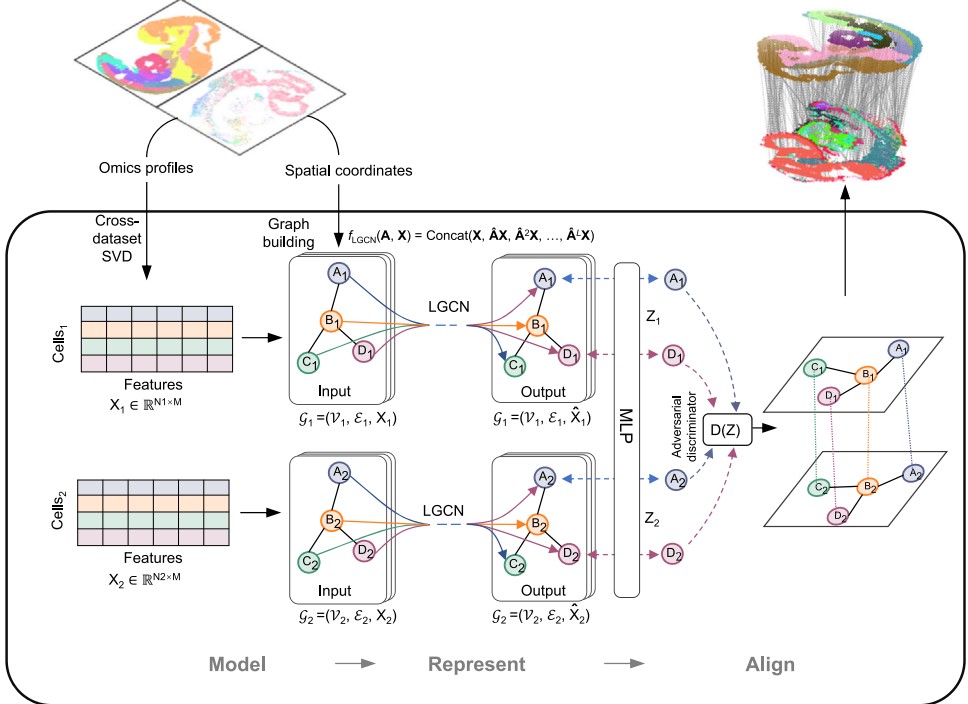

**SLAT: Spatially-Linked Alignment Tool**

**Fig. 1 | Architecture of SLAT framework.** The SLAT algorithm can be divided into three main steps: model, represent and align. We model omics data from two single-cell spatial slices as low dimensional representations $\mathbf{X}_1 \in \mathbb{R}^{N_1 \times M}, \mathbf{X}_2 \in \mathbb{R}^{N_2 \times M}$ obtained via SVD-based cross-dataset decomposition, where $N_1, N_2$ are cell numbers of each slice and $M$ is the SVD dimensionality. We use spatial coordinates of the cells to build $K$-neighbor spatial graphs $\mathscr{G}_1 = (\mathscr{V}_1, \mathscr{E}_1, \mathbf{X}_1), \mathscr{G}_2 = (\mathscr{V}_2, \mathscr{E}_2, \mathbf{X}_2)$ in each slice, respectively. The neighbor size can be set dynamically to adjust for uneven spatial resolution across technologies. To encode local and global spatial information, SLAT uses $L$-layer lightweight graph convolution networks to propagate and aggregate information across the spatial graph into cell embeddings $\hat{\mathbf{X}}_1, \hat{\mathbf{X}}_2$, which are then projected into alignment space $\mathbf{Z}_1, \mathbf{Z}_2$ with a multi-layer perceptron trained adversarially against a discriminator $f_D(\mathbf{Z})$ to align the two graphs by minimizing their Wasserstein distance (see "Methods"). Node labels with the same letters (e.g., $A_1, A_2$) represent corresponding cells in different slices.

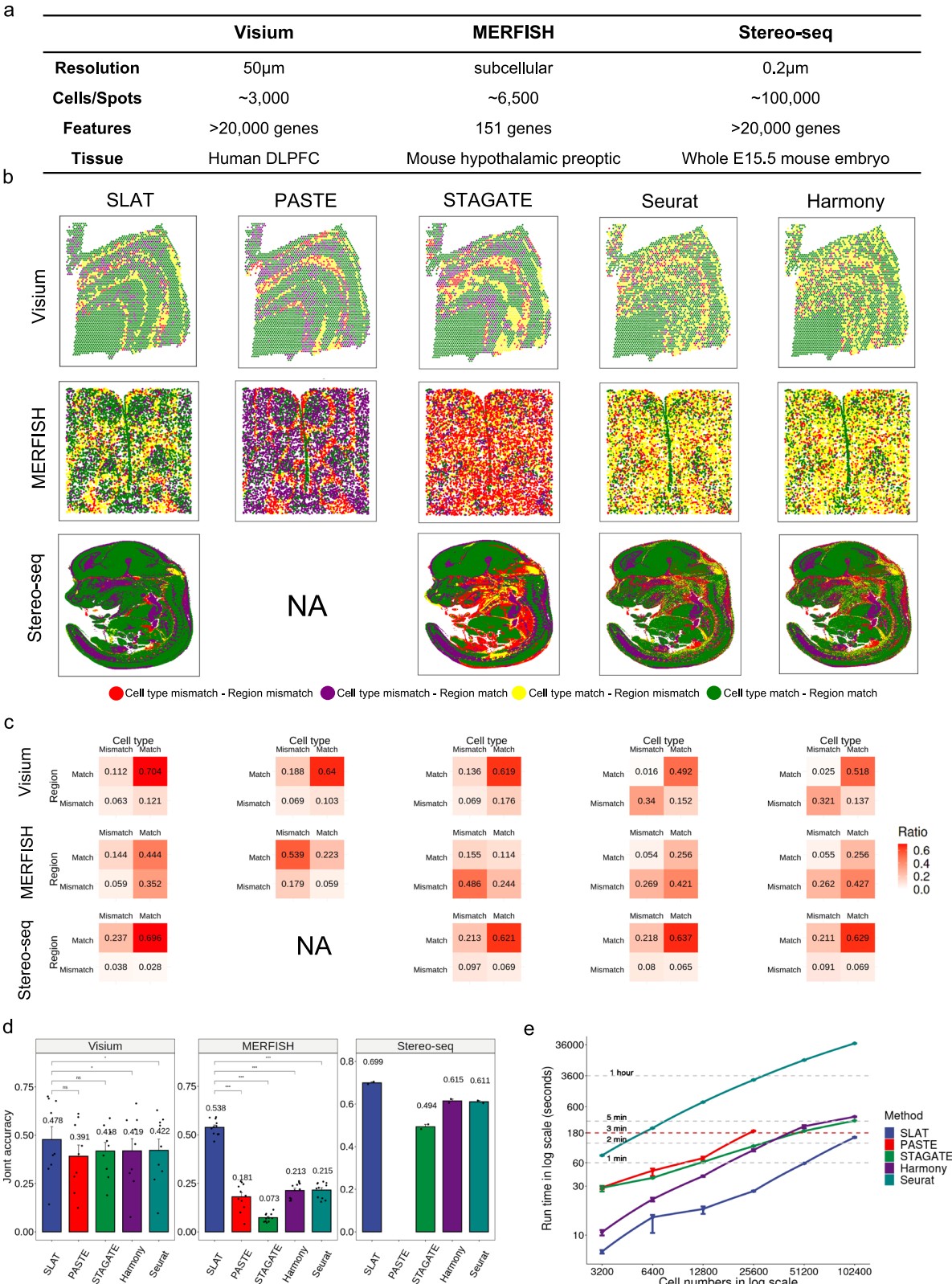

Meanwhile, we noticed a gap between cell type macro and micro F1 scores (by ~0.15) in the MERFISH and Stereo-seq datasets for all methods (Supplementary Fig. 8), which is largely due to mismatches in rare cell types (Supplementary Fig. 10). E.g., "OD mature 3" (6 cells) and "OD mature 4" (10 cells) in the MERFISH dataset tend to be aligned to "OD mature 2", a transcriptionally similar cell type, by SLAT. Similar matchings were also produced by Seurat and Harmony, while PASTE

and STAGATE incorrectly aligned most "OD mature 3" and "OD mature 4" cells with "Inhibitory," "Endothelial," or "Ambiguous." Transcriptionally more distinct rare cell types such as "Pericytes" and "Microglia" were well-aligned by SLAT, Harmony and Seurat, but misaligned by spatially aware algorithms PASTE and STAGATE, suggesting that SLAT's combination of graph convolution and adversarial learning could integrate spatial context and molecular features more

**Fig. 2 | Evaluation on homogeneous spatial alignment. a** Summary of the three benchmark datasets. **b** Visualization of alignment results of different methods on the benchmark datasets in (**a**). Slices are colored according to alignment correctness of cell type and spatial region. For example, green means both cell type and region are corrected aligned. Only alignments between the first two slices are shown. PASTE failed to run on the Stereo-seq dataset due to GPU memory overflow (capping at 80 GB). **c** Heatmaps quantifying the region matching accuracy and cell-type matching accuracy, respectively, in (**b**), in the form of contingency tables. The number in each cell is the average proportion across eight repeats with different random seeds. **d** Joint accuracy ("Methods") of different methods on the aggregated datasets. PASTE failed to run on the Stereo-seq dataset due to GPU memory overflow (capping at 80 GB). $n$ = 9, 11 and 2 for Visium, MERFISH and Stereo-seq datasets, respectively. Error bars indicate mean ± s.d. $p$ values were calculated using the two-sided paired Wilcoxon rank sum test for Visium and MERFISH datasets. The $p$ values of SLAT against PASTE, STAGATE, Harmony, Seurat are 0.16, 0.13, 0.02, 0.02 in Visium dataset (left panel), and $9.8 \times 10^{-4}$, $9.8 \times 10^{-4}$, $9.8 \times 10^{-4}$, $9.8 \times 10^{-4}$ in MERFISH dataset (middle panel). *$p < 0.05$; ***$p < 0.001$; ns $p \geq 0.05$. **e** Running time of each method on subsampled datasets of varying sizes. $n$ = 8 repeats with different subsampling random seeds. Error bars indicate mean ± s.d. Source data are provided as a Source Data file.

effectively. On the other hand, mismatches in the Stereo-seq dataset could mostly be attributed to inconsistent taxonomies in the source annotation, e.g., cells annotated as various organs were partially aligned with "connective tissue," which does exist in most organs.

SLAT's lightweight graph convolutional component effectively improves model robustness. Evaluation with subsampled data of various sizes (spanning from 200 to 102,400) showed that SLAT consistently provides the best results, even with as few as 200 cells (Supplementary Fig. 11, "Methods"). Further inspection confirmed that the performance of SLAT remains highly robust to a wide range of hyperparameter settings and against random corruption of the spatial graph (Supplementary Fig. 12a, b, "Methods").

As technologies continue to evolve, the throughput of spatial single-cell experiments is constantly increasing[19]. Implemented as a neural network and optimized for parallelism, SLAT is highly scalable: benchmarks showed that SLAT is consistently the fastest method across all data sizes and aligns slices each with over 100,000 cells in just 3 min (Fig. 2e). We also noticed that PASTE fails to run as cell number exceeds 25,600, partly due to its memory intensive implementation for optimal transport.

## Matching heterogeneous datasets across distinct technologies and modalities

Spatial alignment across multiple technologies and modalities amalgamates complementary information towards a comprehensive in situ view of cellular states[20]. Benefiting from its unique design, SLAT provides reliable alignment across heterogeneous datasets, which is largely beyond the capability of current alignment algorithms.

We first used SLAT to align spatial transcriptomic datasets generated by two distinct technologies with varying scale and detectability[8,21]. Specifically, the seqFISH dataset contains over 10,000 cells with only 350 genes detected in total[3,21], whereas the Stereo-seq dataset contains about 5,000 cells covering over 20,000 genes[8]. Meanwhile, the cell types in the two datasets were annotated at different resolutions: the seqFISH dataset has finer-grained annotation (21 cell types, Supplementary Fig. 13a) than the Stereo-seq dataset (11 cell types, Supplementary Fig. 13b). The high-quality alignment produced by SLAT (Fig. 3a) enables an accurate label transfer for cell typing improvement (Supplementary Fig. 13c). For example, cells labeled as "Neural crest" in the Stereo-seq dataset were aligned to seqFISH regions with four major cell types (Fig. 3b and Supplementary Fig. 13d), refining cell typing which was further validated by known marker genes (Fig. 3c and Supplementary Fig. 13e). Notably, slices in the two datasets also differ in spatial structure with substantial non-rigid deformations (Supplementary Fig. 13a, b). Such challenging scenario effectively failed methods other than SLAT (Supplementary Figs. 14 and 15a). Similar cross-scale alignment can also be achieved on Visium and Xenium slices[4], where SLAT accurately pinpoints a rare group of triple positive breast tumor cells, while all other methods failed (Supplementary Fig. 16).

More challenging is the cross-modality spatial alignment, which is partly due to the disjoint feature spaces that invalidate the use of our SVD-based cross dataset matrix decomposition strategy as well as other canonical batch correction methods. Benefiting from the modular design of SLAT (Fig. 1), we employed the graph-linked multi-modality embedding strategy we proposed before[22] to project cells of different modalities into a shared embedding space before feeding them into the LGCNs ("Methods"). With this extension, SLAT successfully produced a spatial alignment across RNA (Stereo-seq) and ATAC (spatial-ATAC-seq) slices. While the different modalities featured drastically different spatial resolutions (0.2 μm for Stereo-seq but 20 μm for spatial-ATAC-seq), SLAT managed to align them well (Fig. 3d and Supplementary Fig. 17a). Cell-type labels transferred from Stereo-seq to spatial-ATAC-seq based on the alignment were consistent with anatomical features and with the accessibility of tissue-specific genes (Fig. 3e, f and Supplementary Fig. 17b). We also experimented with other spatial alignment methods by feeding the same multi-modality embeddings as input, but found suboptimal results (Supplementary Fig. 15b and 18). Joint regulatory analysis using the aligned cell pairs and transferred labels effectively identified various key regulators in the heart, such as *Jund* and *Ctnnb1*[23,24] ("Methods," Supplementary Fig. 17c), which could not be identified using the spatially unaware GLUE embeddings (Supplementary Fig. 19).

## Mapping fine-scale spatial-temporal transitions by developmental alignment

Embryonic development is a highly dynamic process with extensive spatial-temporal transitions involving the generation, maturation, and functional alteration of organs and tissues at specific timepoints. To probe spatial-temporal dynamics during early development, we used SLAT to align two spatial atlases of mouse embryonic development at E11.5 and E12.5[8] (Fig. 4a).

Most cells in the brain, heart, and liver were well aligned with high alignment similarity scores, consistent with their spatial-temporal conservation at the two timepoints (Fig. 4b and Supplementary Fig. 20a–c). In addition, several regions were enriched with less-aligned cells of lower alignment similarity scores, which may be attributed to both biological development (e.g., the newly emerged organs kidney and ovary at region I[8,25–27], the rapid enlargement of lung primordium and its displacement from the upper part of the heart to the lower part at region II[28] as shown in Supplementary Fig. 20d, and the disappearance of branchial arch at corresponding position of region III in E11.5[8]) and technical variation (e.g., tissue loss at corresponding position of region IV in E11.5).

We next followed up on kidney and ovary, two newly emerging organs at around E11.5–12.5[8,25–27]. SLAT accurately identified them both as developing from the "Urogenital ridge" at E11.5 (highlighted by the green box in Fig. 4c). Consistent with previous reports, SLAT alignment showed precisely that the ovary develops directly from a single area[29] (left panel of Fig. 4d) while the kidney develops from two separate areas corresponding to the mesonephros and metanephros structures in early kidney development[25,30,31] (right panel of Fig. 4d and Supplementary Fig. 20e). In addition, based on the clustering analysis of aligned cells, we further identified a group of rare nephron progenitor cells located in the mesonephros, likely corresponding to an ephemeral mesonephric tubule during mesonephros

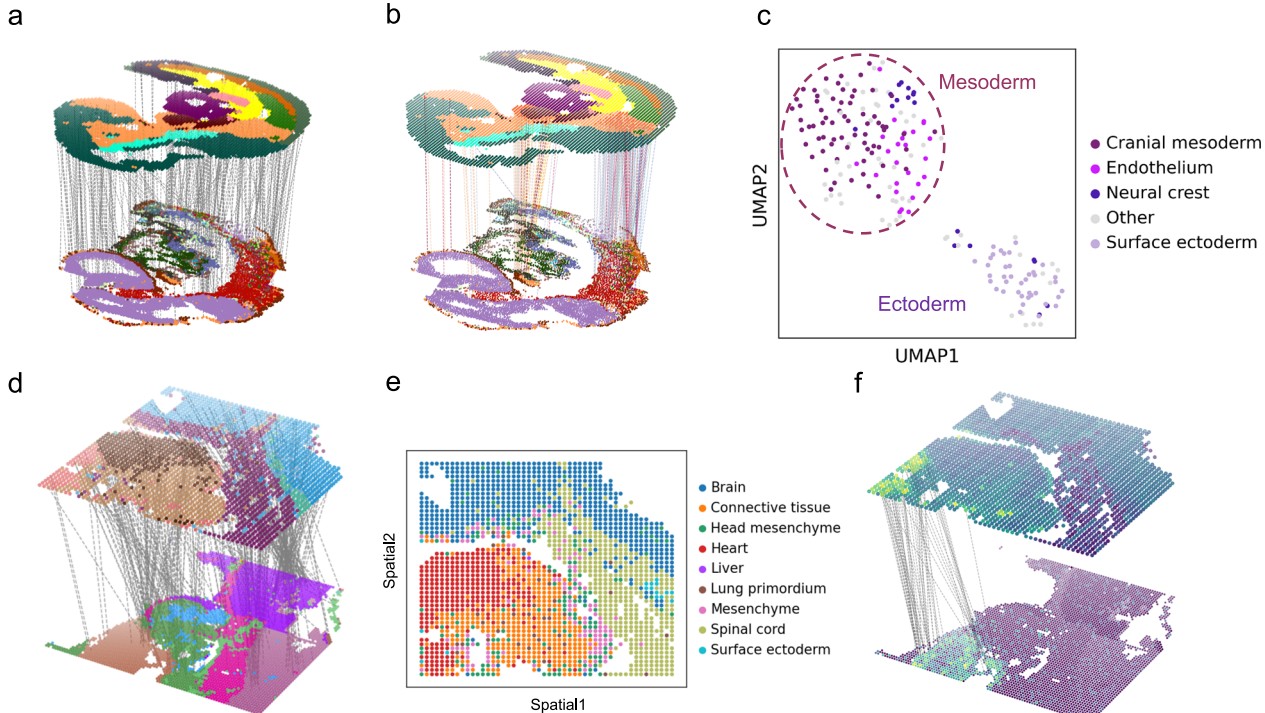

**Fig. 3 | Spatial alignment across distinct technologies and modalities.**
**a** Visualization of the alignment of E8.75 seqFISH mouse embryo and E9.5 Stereo-seq mouse embryo datasets, colored by cell types (subsampled to 300 alignment pairs for clear visualization). **b** Highlighting the alignment of cells labeled as "Neural crest" in Stereo-seq (top) to seqFISH (bottom), alignment lines are colored by cell type of aligned cells in seqFISH. One of the cells in the head region of Stereo-seq that aligned to the abdomen region of seqFISH is a neural crest cell which migrates throughout the whole embryo. **c** UMAP visualization of cells labeled as "Neural crest" in Stereo-seq, colored by cell type of aligned cells in seqFISH, cell types with

fractions less than 5% are collectively labeled as "Other". Dashed circles indicate manual annotation by marker genes (corresponding to Supplementary Fig. 7d, e). **d** Visualization of the alignment of E11.5 spatial-ATAC-seq mouse embryo and E11.5 Stereo-seq mouse embryo datasets, with spatial-ATAC-seq dataset colored by clusters and Stereo-seq colored by cell types (subsampled to 300 alignment pairs for clear visualization). **e** Cell-type labels of spatial-ATAC-seq transferred from Stereo-seq via SLAT alignment. **f** Showing chromatin accessibility score and gene expression pattern of heart marker *Tnnt2* on SLAT alignment.

development[32] (Fig. 4e, f). Interestingly, the results show that the mesonephros is spatially adjacent to the ovary, which is consistent with the well-documented developmental cascade whereby a part of the mesonephros develops into the fallopian tube after degeneration (Fig. 4d). Our findings revealed the unique value of heterogeneous spatial alignment for interrogating the spatial-temporal process of organogenesis.

We also attempted the same alignment task with other methods (Supplementary Fig. 21). The spatially unaware algorithms (Seurat, Harmony) simply missed mesonephros, while the spatially aware algorithms (PASTE, STAGATE) align them to incorrect cells. Of interest, SLAT's unique capability on heterogeneous spatial alignment further enables spatiotemporally progressive alignment across multiple development stages: applying SLAT to mouse embryo slices spanning E9.5–E16.5 effectively recapitulated the development dynamics of multiple organs across all three germ layers[8] (Supplementary Fig. 22).

## Discussion

One of the essential challenges in spatial omics alignment is to appropriately model spatial context. Early methods such as Splotch[13] and Eggplant[33] model slices as rigid bodies and require manual annotation of landmark spots to guide the alignment. PASTE[14] eliminates the need for landmark annotation by considering gene expression of all spatial spots, but its reliance on exact spatial distance impedes application to heterogeneous alignments involving complex non-rigid structural alterations. By combining spatial graph convolution and adversarial matching, SLAT achieves reliable spatial alignment for both

homogeneous and heterogeneous slices in an unsupervised, data-oriented manner.

3D reconstruction from consecutive slices is a common application of spatial alignment. SLAT supports reconstruction from multiple slices via progressive pairwise alignment (e.g., Supplementary Fig. 23 for a consecutive stack of four slices from the same E15.5 mouse embryo). While similar 3D stacking can also be achieved with PASTE[14], SLAT is better equipped to account for non-rigid structural shifting and alteration among slices, enabling adaptive correction for potential deformation artifacts (see Supplementary Fig. 24 for a quantitative assessment).

We noticed that other topology-based metrics have been proposed as performance measurements, such as the edge score which quantifies preservation of spatial neighbors[34] (see Supplementary Fig. 25a and "Methods"). However, we'd argue that these metrics could be intrinsically biased as they measure the continuity of matching solely which could be particularly problematic with the existence of structural changes (see Supplementary Fig. 25b and "Methods" for a counterexample).

SLAT's unique ability to conduct heterogeneous spatial alignments promises a wide range of biological applications. In particular, such spatial alignment enables identifying spatially-resolved changes such as key alterations of spatial patterns during development. Meanwhile, aligning slices from the same tissue generated by different technologies could enable in silico data enhancement, and ultimately combining their complementary advantages in spatial resolution and genomic coverage. Meanwhile, proper cross-modal alignment further sheds lights on key regulators and corresponding regulatory circuits.

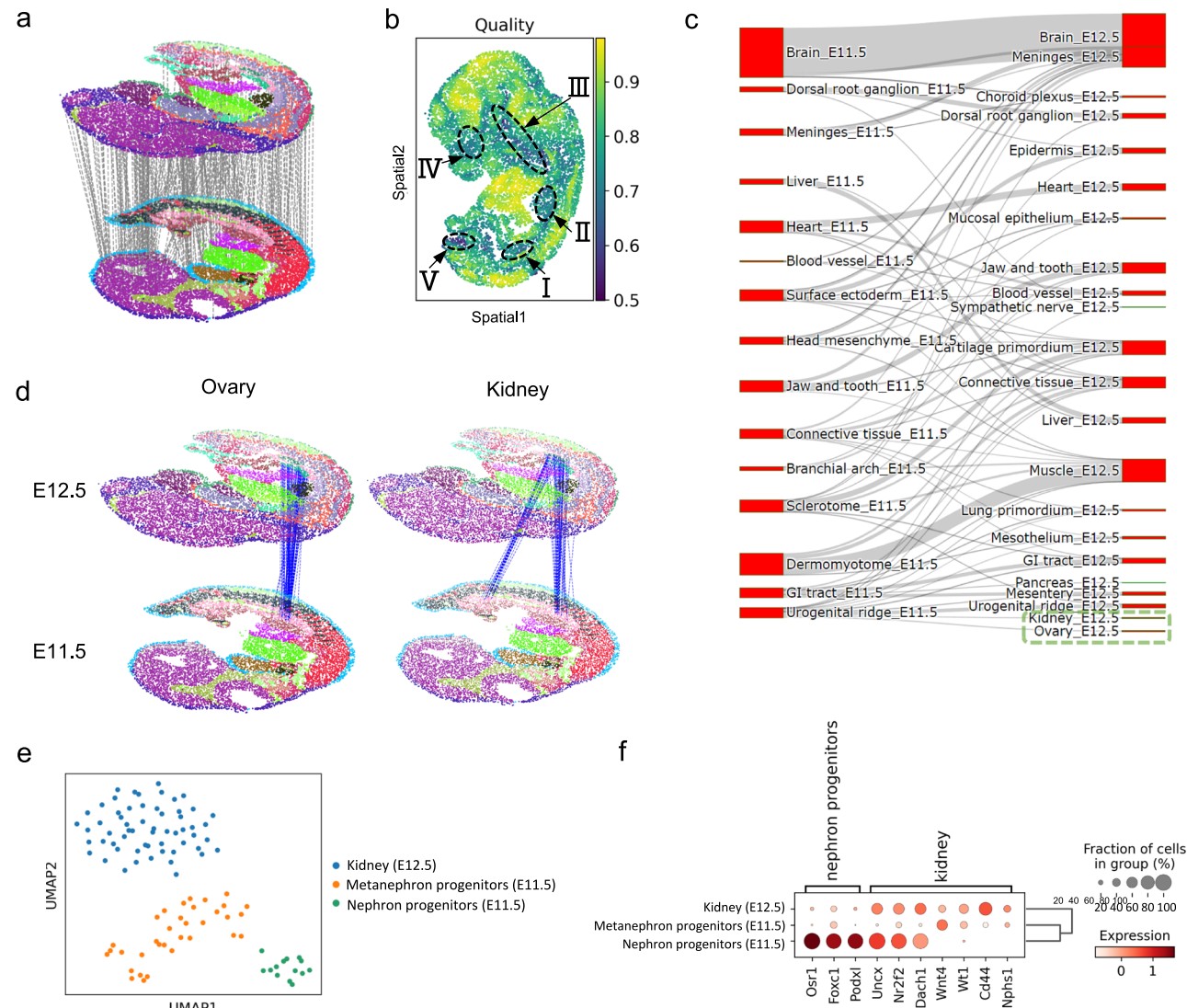

**Fig. 4 | Developmental alignment of mouse embryo. a** Visualization of the alignment of E11.5 and E12.5 mouse embryo Stereo-seq datasets, colored by cell types (subsampled to 300 alignment pairs for clear visualization). **b** Similarity score of the alignment. Higher scores indicate higher alignment confidence. Dashed circles highlight five regions with low similarity scores. **c** Sanky plot showing cell type correspondence of SLAT alignment between the two datasets. The green box highlights cells labeled as "Kidney" and "Ovary" in E12.5. **d** Alignment visualization highlighting cells labeled as "Ovary" (left) and "Kidney" (right) in E12.5 and their aligned cells in E11.5, respectively. **e** UMAP visualization of cell type annotations for cells labeled as "Kidney" in E12.5 and their aligned cells in E11.5. **f** Dot plot showing marker gene expression of cell types in (**e**). Source data are provided as a Source data file.

SLAT is fast. In fact, generating the input cell embeddings is the most time-consuming step, while training and inferring with the SLAT core model takes only about 10 s for $10^6$ cells. Once the input embeddings are ready, aligning millions of cells can be completed in near-real-time, enabling an efficient search of a massive database within an affordable timeframe. SLAT's blazing speed sets the stage for some exciting applications such as inferring spatially dependent causal mechanisms by systematically comparing multiple Perturb-map slices[35], or constructing whole organ 3D atlases involving thousands of slices[36].

Last but not least, designed as a flexible framework, SLAT can be readily adapted and extended. For instance, additional information such as expert curation may be incorporated into the coordinate matching module to help distinguish symmetric structures and polish the final alignment ("Methods"). Meanwhile, the spatial graph modeling technique employed in SLAT may also be adapted to address other problems, such as comparative alignment across species.

Overall, SLAT provides a unified framework for various spatial integration scenarios. To promote its application by the research community, the SLAT package, along with detailed tutorials and demo cases, is available online at https://github.com/gao-lab/SLAT.

## Methods

### SLAT framework

**Joint modeling of spatial coordinates and omics features.** We denote a spatial omics dataset as $\mathscr{D} = \left\{ \left( \mathbf{g}^{(i)}, \mathbf{s}^{(i)} \right), i = 1, 2, \ldots, N \right\}$, where $N$ is the number of spots or cells, $\mathbf{g}^{(i)} \in \mathbb{R}^G$, and $\mathbf{s}^{(i)} \in \mathbb{R}^2$ are the raw omics features (e.g., genes) and spatial coordinates of cell $i$, respectively, where $G$ is the number of omics features. For datasets containing non-identical omics features, we use their overlapping features. For ease of notation, we denote the combination of omics features and spatial coordinates across all cells in a dataset as matrices $\mathbf{G} \in \mathbb{R}^{N \times G}$, $\mathbf{S} \in \mathbb{R}^{N \times 2}$, respectively. Subscripts such as $\mathbf{G}_1, \mathbf{G}_2$ and $\mathbf{S}_1, \mathbf{S}_2$ are added to distinguish two datasets being aligned.

In attempts to correct inter-sample batch effects, we employ an SVD-based cross-dataset matrix decomposition strategy as a preprocessing step. To begin with, we denote the log-normalized and scaled omics matrices of two spatial datasets as $\widetilde{\mathbf{G}}_1 \in \mathbb{R}^{N_1 \times G}$ and

$\widetilde{\mathbf{G}}_2 \in \mathbb{R}^{N_2 \times G}$. We then apply SVD on their dot product as follows:

$$\widetilde{\mathbf{G}}_1 \widetilde{\mathbf{G}}_2^\top = \mathbf{U}\boldsymbol{\Sigma}\mathbf{V}^\top \tag{1}$$

Using the decomposed matrices, we obtain the batch-corrected embeddings of the two datasets as:

$$\begin{aligned}\mathbf{X}_1 &= \mathbf{U}_{1:M}\boldsymbol{\Sigma}_{1:M}^{\frac{1}{2}}\\ \mathbf{X}_2 &= \mathbf{V}_{1:M}\boldsymbol{\Sigma}_{1:M}^{\frac{1}{2}}\end{aligned} \tag{2}$$

where $\mathbf{U}_{1:M} \in \mathbb{R}^{N_1 \times M}$, $\mathbf{V}_{1:M} \in \mathbb{R}^{N_2 \times M}$, $\boldsymbol{\Sigma}_{1:M} \in \mathbb{R}^{M \times M}$ are truncated decompositions corresponding to the $M$-largest singular values.

The SVD-based correction performed well with comparable (and somewhat superior) joint accuracy over state-of-the-art method Harmony[18,37] (Supplementary Fig. 12c). Notably, this preprocessing step is also flexible and allows for modular input (e.g., embedding produced by dedicated algorithms like GLUE[22] for cross-modality integration).

We model the spatial information of cells in $\mathscr{D}$ as a spatial graph $\mathscr{G} = (\mathscr{V}, \mathscr{E}, \mathbf{X})$, where each node $v_i \in \mathscr{V}$ corresponds to a cell with the batch-corrected embedding $\mathbf{x}^{(i)} \mathbb{R}^M$ as its node attribute, and the edges connect the $K$-nearest neighbors in the spatial space. We also denote the adjacency matrix of the graph as $\mathbf{A} \in \{0,1\}^{N \times N}$. $\mathbf{A}_{i,j} = 1$ if the edge $(v_i, v_j) \in \mathscr{E}$, otherwise $\mathbf{A}_{i,j} = 0$. In particular, for cross-technology alignment, different technologies may have distinct spatial resolutions, in which case we can select different $K$'s for each slice according to its spatial resolution. SLAT also supports building the spatial graph by radius, where all cells located within a specific radius are taken as neighbors.

SLAT formulates the alignment of two spatial datasets $\mathscr{D}_1 = \left\{ \left(\mathbf{x}_1^{(i)}, \mathbf{s}_1^{(i)}\right), i=1,2,\ldots,N_1 \right\}$ and $\mathscr{D}_2 = \left\{ \left(\mathbf{x}_2^{(i)}, \mathbf{s}_2^{(i)}\right), i=1,2,\ldots,N_2 \right\}$ as a minimum-cost bipartite matching problem of their corresponding spatial graphs $\mathscr{G}_1 = (\mathscr{V}_1, \mathscr{E}_1, \mathbf{X}_1)$ and $\mathscr{G}_2 = (\mathscr{V}_2, \mathscr{E}_2, \mathbf{X}_2)$:

$$\min_{\mathscr{M}} \sum_{(v_i, v_j) \in \mathscr{M}} \|\mathbf{z}_1^{(i)} - \mathbf{z}_2^{(j)}\|, \text{s.t. } \mathscr{M} \subset \mathscr{V}_1 \times \mathscr{V}_2, |\mathscr{M}| = m, \tag{3}$$

where $\mathbf{z}_1^{(i)}, \mathbf{z}_2^{(j)} \in \mathbb{R}^P$ are node embeddings of $v_i$ and $v_j$ in $\mathscr{G}_1$ and $\mathscr{G}_2$, respectively, $\mathscr{M}$ is a set of matches of fixed size, and $P$ is the dimensionality of node embeddings. Similarly, we use matrices $\mathbf{Z}_1 \in \mathbb{R}^{N_1 \times P}$ and $\mathbf{Z}_2 \in \mathbb{R}^{N_2 \times P}$ to denote the combination of cell embeddings of all cells in the two datasets.

It has been demonstrated in a previous work, that the above matching problem is equivalent to minimizing the Wasserstein distance between node embeddings from different graphs[16]. SLAT follows the same approach with adaptations for spatial omics data. Below, we explain how node or cell embeddings $\mathbf{Z}_1, \mathbf{Z}_2$ can be obtained and optimized for spatial graph alignment.

**Construction of holistic cell representations.** An accurate alignment of spatial omics datasets should align cells that are similar in both the molecular modality and the spatial context. In particular, the spatial context can involve various resolutions, ranging from microenvironments to global positions within the tissue. Inspired by previous work[16,38,39], we first employ the lightweight graph-convolutional network (LGCN) to derive a holistic cell representation with all such information integrated for each dataset. A LGCN propagates and aggregates information along the spatial graph through stepwise concatenations:

$$\widetilde{\mathbf{X}} = f_{\text{LGCN}}(\mathbf{A}, \mathbf{X}) = \text{Concat}\left(\mathbf{X}, \hat{\mathbf{A}}\mathbf{X}, \hat{\mathbf{A}}^2\mathbf{X}, \ldots, \hat{\mathbf{A}}^L\mathbf{X}\right), \tag{4}$$

where $\hat{\mathbf{A}} = \widetilde{\mathbf{D}}^{-\frac{1}{2}}\widetilde{\mathbf{A}}\widetilde{\mathbf{D}}^{-\frac{1}{2}}$, $\widetilde{\mathbf{A}} = \mathbf{A} + \mathbf{I}$, $\widetilde{\mathbf{D}}$ is the diagonal degree matrix of $\widetilde{\mathbf{A}}$, and $L$ is the maximal number of steps. The resulting cell representation

$\widetilde{\mathbf{X}} \in \mathbb{R}^{N \times (L+1)M}$ is a concatenation of multi-level information. The first $M$ dimensions correspond to no graph propagation, which is simply a copy of the omics data $\mathbf{X}$. The second $M$ dimensions correspond to one-step graph propagation, reflecting the composition of a cell's immediate neighbors, which form its microenvironment. The information coarsens as the number of steps increases, gradually becoming a representation of rough locations within the tissue. Thus, $\widetilde{\mathbf{X}}$ contains informative features for spatial alignment at multiple levels of spatial context.

The cell representation $\widetilde{\mathbf{X}}$ is constructed separately for each dataset by using dataset-specific omics profiles and adjacency matrices:

$$\begin{aligned}\widetilde{\mathbf{X}}_1 &= f_{\text{LGCN}}(\mathbf{A}_1, \mathbf{X}_1),\\ \widetilde{\mathbf{X}}_2 &= f_{\text{LGCN}}(\mathbf{A}_2, \mathbf{X}_2).\end{aligned} \tag{5}$$

**Adversarial graph alignment.** Based on the holistic cell representations $\widetilde{\mathbf{X}}_1, \widetilde{\mathbf{X}}_2$ described above, we use adversarial alignment to learn cell embeddings $\mathbf{Z}_1, \mathbf{Z}_2$ that minimize the Wasserstein distance for graph matching[16].

Specifically, we apply a multilayer perceptron denoted as $f_Z$ to mitigate systematic bias in $\widetilde{\mathbf{X}}_1$ and $\widetilde{\mathbf{X}}_2$ that may arise from differences in omics distribution or spatial topology across datasets:

$$\begin{aligned}\mathbf{Z}_1 &= f_Z\left(\widetilde{\mathbf{X}}_1\right),\\ \mathbf{Z}_2 &= f_Z\left(\widetilde{\mathbf{X}}_2\right).\end{aligned} \tag{6}$$

We then introduce the Wasserstein discriminator $f_D$, which uses $\mathbf{Z}_1, \mathbf{Z}_2$ as input and tries to maximize the following Wasserstein loss $L_W$ to estimate the Wasserstein distance:

$$L_W = \frac{1}{|\mathscr{V}_1|}\sum_{v_i \in \mathscr{V}_1} f_D\left(\mathbf{z}_1^{(i)}\right) - \frac{1}{|\mathscr{V}_2|}\sum_{v_j \in \mathscr{V}_2} f_D\left(\mathbf{z}_2^{(j)}\right), \tag{7}$$

where $\mathbf{z}_1^{(i)}$ is a row in $\mathbf{Z}_1$ corresponding to cell $v_i$, and $\mathbf{z}_2^{(j)}$ is a row in $\mathbf{Z}_2$ corresponding to cell $v_j$. The transformation $f_Z$ can then be adversarially trained to minimize (7), for aligning the distribution of cell embeddings in the two datasets properly. However, different single-cell spatial datasets may contain different cell-type proportions or distinct spatial regions. It is thus unreasonable to assume identical distribution of their cell embeddings as assumed in the standard scheme described above[40]. Inspired by a previous study[16], we use the output of the Wasserstein discriminator $f_D$ as a dynamic clipping criterium to select $c \times N_1$ and $c \times N_2$ cells from the two datasets with minimum Wasserstein distance for adversarial training (Supplementary Fig. 1):

$$\begin{aligned}\mathscr{V}_1' &= \arg k \min_{v_i \in \mathscr{V}_1' \subseteq \mathscr{V}_1} f_D\left(\mathbf{z}_1^{(i)}\right), |\mathscr{V}_1'| = c \times N_1,\\ \mathscr{V}_2' &= \arg k \max_{v_j \in \mathscr{V}_2' \subseteq \mathscr{V}_2} f_D\left(\mathbf{z}_2^{(j)}\right), |\mathscr{V}_2'| = c \times N_2,\end{aligned} \tag{8}$$

where $c$ is a hyperparameter between 0 and 1. These cells correspond to the most reliable anchors to guide the alignment. The Wasserstein discriminator loss $L_W$ is then modified accordingly as follows:

$$L_W = \frac{1}{|\mathscr{V}_1'|}\sum_{v_i \in \mathscr{V}_1'} f_D\left(\mathbf{z}_1^{(i)}\right) - \frac{1}{|\mathscr{V}_2'|}\sum_{v_j \in \mathscr{V}_2'} f_D\left(\mathbf{z}_2^{(j)}\right). \tag{9}$$

This approach ensures that distinct regions across two spatial datasets will not be forcibly aligned. The results show that SLAT performs best when $c = 0.6$, although the performance depends only weakly on $c$ (Supplementary Fig. 12a).

To avoid a degenerate solution where all embeddings collapse to a singular point, we adopt an additional reconstruction term to ensure that the embeddings have sufficient information to reconstruct input, essentially enhancing model stability. We use a simple multilayer perceptron network denoted as $f_R$ for data reconstruction, making the following reconstruction loss:

$$L_R = \frac{1}{|\mathcal{V}_1|} \sum_{v_i \in \mathcal{V}_1} \left\| f_R\left(\mathbf{z}_1^{(i)}\right) - \mathbf{x}_1^{(i)} \right\| + \frac{1}{|\mathcal{V}_2|} \sum_{v_j \in \mathcal{V}_2} \left\| f_R\left(\mathbf{z}_2^{(j)}\right) - \mathbf{x}_2^{(j)} \right\|. \quad (10)$$

We assessed the necessity of Wasserstein discriminator though an ablation study on the homogenous (Stereo-seq) and heterogenous (spatial-ATAC-seq vs. Stereo-seq) alignments, respectively. We found that ablation of Wasserstein discriminator did not significantly affect accuracy in homogeneous alignments where difference in the spatial domain is negligible (Supplementary Table 2). However, it did substantially influence accuracy in the heterogeneous alignment of Stereo-seq (0.22 μm) vs. spatial-ATAC-seq (20 μm) where the spatial domain is drastically different in resolution (Supplementary Table 3).

**Overall objective.** Finally, the overall training objective of SLAT can be summarized as follows:

$$\max_{f_D} \alpha \cdot L_W, \quad (11)$$

$$\min_{f_Z, f_R} \alpha \cdot L_W + (1 - \alpha) \cdot L_R, \quad (12)$$

where $L_W$ and $L_R$ are defined by Eqs. (9) and (10), respectively, $\alpha$ is a hyperparameter balancing the contribution of adversarial alignment and data reconstruction. We use stochastic gradient descent (SGD) with the Adam optimizer to train the SLAT model.

**Coordinate matching.** Apart from the core model described above, we also provide options to use additional information to match the spatial coordinates $\mathbf{S}$ of two slices which can help distinguish symmetric structures (e.g., left and right hemispheres of the brain) and improve the final matching quality. The goal of coordinate matching is to roughly align different slices in terms of their overall direction by estimating an affine transformation matrix $\mathbf{M}$. The exact strategy depends on the type of information available.

First, guided by expert knowledge of tissue structures, $\mathbf{M}$ can be computed by combining the scaling, rotation, and translation operations required to obtain a rough alignment between the two slices. Second, if imaging data like H·E staining images are available, $\mathbf{M}$ can be estimated following our tutorial based on SimpleElastix[41], which is a state-of-the-art medical image registration tool. Finally, if no other information is available, we also provide a default solution based on iterative closet point (ICP)[42], a point-cloud registration algorithm, where we treat the spatial datasets as point clouds on a two-dimensional plane and uses geometric features for registration. With the obtained $\mathbf{M}$ matrix, we consort the coordinates of the two datasets by the following transformation:

$$\mathbf{S}_2' = \mathbf{S}_2 \cdot \mathbf{M}. \quad (13)$$

**Quality assessment and probabilistic matching.** With the cell embeddings $\mathbf{Z}_1, \mathbf{Z}_2$ learned by the SLAT core model and the matched spatial coordinates $\mathbf{S}_1$ and $\mathbf{S}_2'$, we match dataset $\mathcal{D}_2$ with $\mathcal{D}_1$ using the following strategy: For cell $i$ in dataset $\mathcal{D}_1$, SLAT first selects the closest $K$ cells from $\mathcal{D}_2$ in matched spatial coordinates $\mathbf{S}$ as a candidate set $\mathscr{C}_i$. We then compute the cosine similarity between the embedding of cell $i$ and each candidate cell in $\mathscr{C}_i$, and evaluate their significance by comparing with a null distribution obtained from 1,000 randomly sampled cell pairs. The final match set $\mathscr{M}_i$ consists of cells from the candidate

set with $p$-values less than 0.05:

$$\mathscr{C}_i = \arg k \min_{v_j \in \mathscr{C}_i \subseteq \mathcal{V}_2} \left\| \mathbf{s}_1^{(i)} - \mathbf{s}_2'^{(j)} \right\|, |\mathscr{C}_i| = K, \quad (14)$$

$$p - \text{value}\left(v_i, v_j\right) = P\left(\cos\left(\mathbf{z}_1, \mathbf{z}_2\right) > \cos\left(\mathbf{z}_1^{(i)}, \mathbf{z}_2^{(j)}\right)\right) \quad (15)$$

$$\mathscr{M}_i = \{v_j \in \mathscr{C}_i | p - \text{value}(v_i, v_j) < 0.05\}, \quad (16)$$

where $\mathbf{s}_1^{(i)}$ and $\mathbf{s}_2'^{(j)}$ are rows in the coordinate matrices $\mathbf{S}_1$ and $\mathbf{S}_2'$, respectively. $K$ is the size of spatial neighborhood as described above.

For convenience of 3D visualization, we only plot the alignment with the smallest $p$-value for each cell in 3D plots (Figs. 3 and 4 and Supplementary Figs. 4, 14, 16, 18, 21–23, and 29).

## Systematic benchmarks

**Benchmark datasets.** We selected 10× Visium, MERFISH and Stereo-seq as representative spatial technologies for benchmarking alignment methods. The 10× Visium dataset comes from consecutive slices of human dorsolateral prefrontal cortex, containing about 3,000 spots per slice, each spanning 50 μm with over 20,000 genes detected[43]. The MERFISH dataset comes from consecutive slices of mouse hypothalamic preoptic, with subcellular resolution but only 151 genes detected in total[21]. The Stereo-seq dataset comes from consecutive slices of an E15.5 mouse embryo, containing over 100,000 single cells per slice, divided into 25 cell types with complex spatial organization, and detects over 20,000 genes in total[8]. We used all available slices in these datasets (9 Visium slices, 12 MERFISH slices and 4 Stereo-seq slices), results of the first two slices of each dataset were presented in Fig. 2, while the accuracy statistics reported were aggregated across all slice pairs. Cell type annotation and tissue region segmentation were obtained from the original authors whenever possible. Since the Stereo-seq dataset does not provide tissue segmentation, we segmented the most prominent regions in the embryo, including Brain, Jaw and face, Spinal cord, Heart, Lung, Liver, and Belly under the guidance of an expert in mouse anatomy. The spatial segmentation of the MERFISH dataset is provided in publication figures but not in raw data, so we re-segmented the data as guided by the figures.

**Slice processing.** For each technology, we first removed the cells/spots that are unannotated in both slices, then rotated the second slice with a random angle before feeding to the alignment methods.

**Benchmarked methods.** The benchmarked methods Harmony, PASTE, STAGATE, and SLAT were executed using the Python packages "harmonypy" (v0.0.6), "paste-bio" (v1.3.0), "STAGATE_pyG" (latest commit 8b9c8ef), and "scSLAT" (v0.2.0), respectively, in Python (v3.8). Seurat was executed using the R package "Seurat" (v4.1.1) in R (v4.1.3). For each method, we used the default data preprocessing steps recommended by the original authors, and searched for the best hyperparameters for each method starting from their default settings (Supplementary Fig. 26). For all experiments involving SLAT, we used the default hyperparameters (SVD dimensionality: 30, graph neighbor: 50, LGCN layer: 3, MLP hidden layer dimension: 256, embedding hidden size: 2048, learning rate: 1e-4, dynamic clipping ratio: 0.6) unless otherwise stated.

**Benchmark tasks.** We benchmarked the alignment methods based on the following four tasks: (1) duplicate slice alignment, (2) real world alignment, (3) split slice alignment, and (4) scalability test:

In duplicate slice alignment, we duplicate and add noise to the first slice of each benchmark dataset by sampling from a negative binomial distribution centered at the measured expression count. We set the inverse dispersion parameter to different values to

simulate varying noise levels:

$$NB(x;\mu,\theta) = \frac{\Gamma(x+\theta)}{\Gamma(x)\Gamma(x+1)} \left(\frac{\mu}{\theta+\mu}\right)^x \left(\frac{\theta}{\theta+\mu}\right)^\theta \qquad (17)$$

where $\mu$ and $\theta$ are the mean and inverse dispersion, respectively. The duplicate slices were also rotated by 60° to avoid leaking information in spatial orientation[14].

In real world alignment, the slices to be aligned are different slices in the benchmark datasets produced from the same position of the same tissue[5,8,43] (Supplementary Fig. 2). We also rotate the slices by 60° before feeding to the alignment algorithms. Each algorithm was run eight times with different model random seeds.

In split slice alignment, we split the first slice of each benchmark dataset into two pseudo-slices of equal size by randomly sampling the cells without replacement. Each algorithm was run eight times with different model random seeds.

For scalability benchmark, we randomly subsampled the Stereo-seq dataset used in real world benchmark to a range of cell numbers (3200, 6400, 12800, 25600, 51200, 102400). The subsampling process was repeated eight times with different random seeds.

**Evaluation metrics.** For synthetic tests on duplicated slices, the ground truth one-to-one matching is known. For each cell $i$ in the original slice, we find cell $j^*$ with maximal matching score on the duplicate slice $j^* = \text{argmax}_{v_j \in \mathscr{C}_i} \cos\left(z_1^{(i)}, z_2^{(j)}\right)$ and compute the ground truth accuracy as follows:

$$\text{ground truth accuracy} = \frac{1}{N}\sum_{i=1}^{N} 1_{j^*=i} \qquad (18)$$

where $1_{j^*=i}$ is an indicator function that evaluates to 1 when $j^* = i$ and 0 otherwise.

In real world spatial alignments, the exact ground truth matching does not exist, but intuitively, a proper spatial alignment should align cells matched in both molecular profile and spatial context. Thus, we reported the cell type matching accuracy and spatial region matching accuracy simultaneously in the form of contingency tables. For convenience, we also defined a "joint accuracy" as the proportion of cells with both cell type and spatial region matched correctly, which corresponds to the upper right corner of the contingency table. We also report the micro and macro F1 score of cell type matching and spatial region matching and joint matching, respectively.

For more comprehensive assessment, we also used the following alternative metrics to quantify the quality of alignments. The first is the estimation of rotation angle that corrects artificial rotation. Specifically, based on the spatial alignment of each benchmarked method, we estimated the optimal corrective rotation angle by solving the Procrustes problem[44], which is then compared with the ground truth to calculated the deviation. However, it is worth noting that rotation estimation may not reliably reflect alignment quality when non-rigid deformations exist.

The second is the edge score, which quantifies how well an alignment preserves neighborhoods[34,45]:

$$A_{nm} = \begin{cases} 1, & if\ n' \sim m' \\ -1, & if\ n'm' \end{cases}$$
$$\text{edge score} = \frac{1}{N}\sum_{n=1}^{N}\sum_{m\in\mathcal{N}_{(n)}} A_{nm} \qquad (19)$$

However, we found that the edge score could be deceived in certain situations. For example, see Supplementary Fig. 25b shows two graphs with known ground truth node pairing information. "Alignment 1" is the correct matching (gray lines) and "Alignment 2" has four mismatched pairs (highlighted by red lines), but the two alignments get the same edge score.

**Benchmark workflow.** We used Snakemake (v7.12.0) to manage the whole benchmark workflow. All benchmarked methods were allocated 16 cores of Intel Xeon Platinum 8358 CPU, 128 GB of RAM, and a NVIDIA A100 GPU with 80 GB VRAM by the Slurm workload management system.

**Hyperparameter robustness.** We tested SLAT's robustness to key hyperparameters including: (1) SVD dimension $M$, (2) number of LGCN layers $L$, (3) learning rate of the SLAT model, (4) MLP hidden layer dimension, (5) dimension $P$ of SLAT embedding, (6) dynamic clipping ratio $c$.

We ran SLAT on the same slices as in Fig. 2b. For Stereo-seq slices containing more than 100,000 cells, we randomly subsampled 8,000 cells from each slice to save time. Every experiment was run 8 times with different model random seeds.

In this test, we also demonstrated the advantage of LCGN architecture, especially in MERFISH dataset, where cell type and spatial region are not significantly correlated (Supplementary Fig. 3). We found substantial decrease in joint accuracy without LGCN (Supplementary Fig. 12a), mainly caused by deteriorated spatial region matching (Supplementary Fig. 28).

**Robustness to noise in the spatial graph.** In practice, the spatial graph may be imperfect due to technical limitations. Therefore, we tested SLAT's robustness to graph corruption in the same dataset used in hyperparameter robustness evaluation. Specifically, we randomly masked the edges in the graph by increasing ratios (from 0.1 and 0.9). Every experiment was run 8 times with different masking random seeds.

## Heterogeneous alignment across distinct technologies and modalities

**Visium and Xenium data alignment.** The Visium and Xenium datasets were generated from consecutive slices of the human breast cancer tissue sample[4]. In order to maintain the comparability with the original paper, we chose the exact same slices used in their analysis (see Fig. 4c of ref. 4). We further selected the shared region between the two slices as the original authors reported[4] for follow-up analysis.

Considering that the Xenium slice contains more than 100,642 cells while Visium only contains less than 3841 spots in same physical region, we used different neighbor sizes proportional to cell density ($K = 5$ for Visium, and $K = 130$ for Xenium) when constructing the spatial graphs, in order to ensure that the GCNs have similar spatial receptive fields. SLAT is then run with otherwise default parameters. We selected Visium triple positive spots based on the number of aligned Xenium triple positive cells (Supplementary Fig. 29c). 7 spots with more than 2 aligned cells were chosen. Following the original paper, we did differential gene expression analysis of the SLAT identified and manually curated triple positive spots against all other spots, respectively, using the Scanpy[46] function "scanpy.tl.rank_genes_groups" with parameter "methods='wilcoxon'". For comparison, all methods included in the benchmarks (PASTE, Harmony, Seurat, and Harmony) were applied to the same Visium and Xenium datasets with their default parameters.

For comparison, all methods benchmarked (PASTE, Harmony, Seurat, and STAGATE) were applied to the same datasets with their default parameters (Supplementary Fig. 16).

**SeqFISH data and Stereo-seq data alignment.** For Stereo-seq and seqFISH, we chose the E9.5 slice with the most complete cell type annotation by the original authors (i.e., the slices with the least proportion of unannotated cells), respectively. To align the chosen seqFISH and Stereo-seq slices, we run SLAT using $K = 20$ for the Stereo-seq

slice and $K = 50$ for the seqFISH+ slice to balance cell density, while using default value for other parameters. We next refined cell type annotations in the Stereo-seq dataset based on the higher resolution annotation in seqFISH through label transfer for "Neural crest" cells (Fig. 3b). We also manually annotated "Neural crest" cells for independent validation. Scanpy[46] was used for the analysis following its official tutorial: the data were log-normalized using the functions "sc.pp.normalize_total" and "sc.pp.log1p". Highly variable genes were identified with "sc.pp.highly_variable_genes". The first 50 principal components after PCA ("sc.tl.pca") were used to generate neighborhood graphs ("sc.pp.neighbors") for computing UMAP embeddings ("sc.tl.umap") and Leiden clustering ("sc.tl.leiden"). The "Neural crest" cells were annotated via marker genes of different germ layers (*Foxc2* and *Vcan* for mesoderm, *Msx1*, *Mif*, and *Dlk1* for ectoderm, see Supplementary Fig. 13e).

For comparison, all methods benchmarked (PASTE, STAGATE, Seurat and Harmony) were applied to the same datasets with their default parameters (Supplementary Fig. 14).

**Spatial-ATAC-seq and Stereo-seq data alignment.** For cross-modality alignment, the slices to align were two 11.5-day mouse embryo datasets from Stereo-seq (RNA) and spatial-ATAC-seq (ATAC), respectively. For Stereo-seq, we chose the E11.5 slice with the most complete cell type annotation by the original authors (i.e., the slice with the least proportion of unannotated cells). For spatial-ATAC-seq, we chose the E11.5 slice with the highest spatial resolution (20 μm). Given that the current spatial-ATAC-seq data did not cover the entire embryo due to technical limitations, we extracted the anatomically corresponding regions from the Stereo-seq dataset under expert guidance.

To project cells from different modalities into a shared latent space, we employed the graph-linked multi-modality embedding strategy we proposed before[22], and built the graph with $K = 50$ for Stereo-seq, and $K = 20$ for spatial-ATAC-seq, then run SLAT with default hyperparameters. Based on the outputted alignment, we transferred cell type labels from Stereo-seq to spatial-ATAC-seq which was not annotated, and then applied SCENIC+[47] for joint regulatory inference.

To compare with benchmarked methods (PASTE, Harmony, Seurat and Harmony), we also used the same multi-modality embedding as their input followed by their default pipeline. Exceptions are Seurat and STAGATE, which do not support low dimensional embeddings as input, thus cannot be compared. In addition, we also compared with the original multi-modality embeddings[22] directly (Supplementary Fig. 18).

**Spatial-temporal alignment.** We applied SLAT to align E11.5 and E12.5 heterogeneous mouse embryo Stereo-seq datasets with the default hyperparameters. In order to maintain the comparability with the original paper, we chose the exact same slices used in their analysis (see Fig. 3 of ref. 8). Regions with lower SLAT similarity scores were marked in Fig. 4b. To focus on kidney development, we extracted cells labeled as "Kidney" in E12.5 and their aligned cells in E11.5, then clustered these cells by using the standard Scanpy clustering pipeline mentioned above and annotated them via well-defined kidney markers[31,32] (*Osr1*, *Foxc1* and *Podxl* for nephron progenitors; *Uncx*, *Nr2f2*, *Dach*, *Wt1*, *Nphs1*, and *Cd44* for kidney; see Fig. 4f). To further demonstrate the robustness of SLAT, we rerun the analysis against two additional slices randomly chosen from E11.5 and E12.5, and obtained similar results (Supplementary Fig. 30).

For comparison, all methods benchmarked (PASTE, STAGATE, Seurat and Harmony) were applied to the same datasets with their default parameters (Supplementary Fig. 21).

**Statistics and reproducibility**
No statistical method was used to predetermine the sample size. No data were excluded from the analyses. The experiments were not

randomized. The Investigators were not blinded to allocation during experiments and outcome assessment. More information was provided in the Reporting summary file.

**Reporting summary**
Further information on research design is available in the Nature Portfolio Reporting Summary linked to this article.

## Data availability
All relevant data supporting the key findings of this study were already published and were obtained from public data repositories. The data used in this study are provided in Supplementary Table 1, including publication and downloading URLs. All benchmarking data and Source data are provided with this paper.

## Code availability
The SLAT framework was implemented in the "scSLAT" Python package, which is available at https://github.com/gao-lab/SLAT[48]. For reproducibility, the scripts for all benchmarks were assembled using Snakemake (v7.12.0), which is also available in the above repository.

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

## Acknowledgements

We thank Mr. H. Yang for his helpful guidance on mouse embryonic development and anatomy, as well as Dr. Z. Zhang, Dr. L. Tao, Dr. F. Tang, Dr. X.S. Xie, Dr. C. Li, and Dr. J. Lu at Peking University for their helpful discussions and comments during the study. This work was supported by funds from the National Key Research and Development Program of China (2022ZD0115004), as well as the State Key Laboratory of Protein and Plant Gene Research, the Beijing Advanced Innovation Center for Genomics (ICG) at Peking University, the Changping Laboratory, and the Shaw Foundation Hong Kong Limited. The research of G.G. was supported in part by the National Program for Support of Top-Notch Young Professionals.

## Author contributions

G.G. and Z.-J.C. conceived the study and supervised the research. C.-R.X. and Z.-J.C. designed and implemented the computational framework and conducted benchmarks and case studies with guidance from G.G. X.-M.T. proposed and implemented the matrix decomposition-based batch correction strategy. C.-R.X., Z.-J.C., X.-M.T., and G.G. wrote the manuscript.

## Competing interests

The authors declare no competing interests.
