## [Peer Review File - NEW · Nature Communications]

Spatial-linked alignment tool (SLAT) for aligning heterogenous slicesReviewer #1 (Remarks to the Author):

Review of 'Spatial-linked alignment tool (SLAT) for aligning heterogeneous slices effectively and efficiently'

The authors present a tool for aligning heterogeneous spatial slices using graph neural networks and adversarial techniques. The paper showcases a substantial range of tested datasets and methods. However, the evaluations and utility of the approach - absolutely central to any methods papers - are severely lacking. This is not even just a major problem but substantially undercuts any contribution from the manuscript.

The authors claim superior performance based on 'systematic benchmarks', but the results fall short of supporting that claim. Apart from the runtime, the only evaluation metric provided is shown in Figure 2 and Extended Data Figure 2. These figures present matrices that measure the alignment of region and cell type annotations in two slices from the same dataset. However, these matrices are incorrectly labeled as confusion matrices. To determine which method is superior, the reader must inspect the 300 lines connecting the two slices and interpret the matrices. This approach leaves much to be desired in terms of clarity, rigor, and ease of comparison.

Moreover, the remaining evaluations in the manuscript are even weaker as they lack any specific metrics and rely solely on visually judging the 300 lines connecting the two aligned images. I wonder why the authors did not test how well the alignment corrects the artificial rotation they applied for some of the comparisons. Additionally, metrics such as micro or macro F1 scores or similar for region and cell-type matches may be useful metrics.

Another important aspect to consider is that the evaluations primarily focus on aligning slices within a homogeneous setting, meaning they are from the same dataset. Given the authors' claim of heterogeneous alignment, I had anticipated more alignment across different datasets rather than just different slices or timepoints. Nevertheless, it is commendable that they did align mouse fetal RNA-seq data with ATAC-seq slices, albeit with the need for careful slice selection and manual cropping prior to applying their method. However, once again, we lack clear metrics beyond the visual inspection of the figures depicting connecting lines to assess the quality of alignment. While the cross-modality ability is interesting, it does not appear to be a widely applicable use case that would capture broad readership interest.

Furthermore, a significant point of concern is the existence of free parameters, such as the SVD dimensionality. The dataset-specific values used for these parameters should be provided for each alignment. It is possible that the lower performance of other methods could be attributed to more extensive fine-tuning of the authors' methods compared to the default settings that were used for the other approaches. Extended Figure 2 does address this to some degree with a metric that is not used in the main text, but it seems to have a duplicated subpanel, which is concerning.

Additionally, Figure 2 fails to include testing of the Seurat and Harmony methods, which should have been compared. Moreover, the authors should have considered downsampling datasets to ensure that all algorithms could be evaluated, rather than excluding them due to memory limitations. This is particularly crucial because PASTE (or the newer PASTE2) is the most similar method, yet the authors did not test it in some cases.

Minor

Cell type is not usually considered a 'molecular feature':
"aligning cells that share molecular features (e.g., cell type)"

Reviewer #2 (Remarks to the Author):

Xia et al. proposes SLAT, a computational method for aligning spatial omics datasets. SLAT begins with a batch correction preprocessing step via matrix decomposition. It then uses LGCN and MLP to learn an embedding. The embedding is encouraged to match across the two datasets by the

design of a Wasserstein discriminator. After training, the embedding is used in combination with spatial information for matching. SLAT is benchmarked with other available methods including PASTE, STAGATE, Seurat and Harmony. It is implemented and tested across various datasets, showing its application in integration across platforms and modalities, inference of regulatory relationship and developmental trajectory. There are some questions regarding the method design, benchmark, and concepts.

Major:

1. The batch removal step and the Wasserstein discriminator seem redundant as they both aim to integrate the embedding across conditions. Given the batch effect is removed by the preprocessing and SLAT uses a shared MLP layer, the necessity of the Wasserstein discriminator needs to be evaluated and better positioned, possibly through an ablation study.

2. In benchmarking, SLAT has better accuracy in identifying cells that are correctly matched by both region and cell types. However, it seems that spatial-unaware methods (Seurat and Harmony) have better accuracy in cell type matching alone compared with SLAT in most settings (Supplementary Fig 2, 4). Why counting the cells that are correctly matched in both aspects serves as a better measure of matching than correct cell type matching alone? From my understanding, correct cell type matching is already enough for a number of certain tasks such as cell type label transfer.

3. There are other methods for matching across samples. For example, Mixscape [1] is a very simple method that is used to match cells in scRNA samples by its nearest neighbors in the PCA space. Mixscape can serve as a baseline for using spatial-unaware embedding to substitute the embedding learned by SLAT optimization. This can also serve as an ablation study to show how the LGCN architecture helps for accurate matching.

4. One-to-one or one-to-many. Is it really reasonable to establish a one-to-one matching while the measured data is noisy and especially not of the same resolution? For example in Mixscape [1], it is suggested to use the mean of nearest neighbors to represent the "counterfactual state". Tangram [2], a method for aligning scRNA-seq and pixel-resolution spatial omics data, also considers a probabilistic matching scheme. In general one-to-many matching may be a more reasonable task to denoise the matching itself and meaningfully generalize to matching across technologies with different resolutions. Although this is mentioned in the discussion part, the issue still needs to be better addressed due to its fundamental importance to the methodology.

Minor:

1. Typo in line 700. Harmony is mentioned twice. Should be STAGATE instead.

2. Supplementary Figure 4 should be better highlighted in the main text.

3. The rationale for using the matrix decomposition based batch correction instead of other batch correction methods needs to be better elaborated.

4. The evaluation metric used is only a minimum requirement for correct matching.

5. When performing benchmarking, the other methods are tested with their default parameters. In cases where the performance is close, will fine-tuning other available methods yield a better result than SLAT?

[1] Papalexi, Efthymia, et al. "Characterizing the molecular regulation of inhibitory immune checkpoints with multimodal single-cell screens." *Nature genetics* 53.3 (2021): 322-331.

[2] Biancalani, Tommaso, et al. "Deep learning and alignment of spatially resolved single-cell transcriptomes with Tangram." *Nature methods* 18.11 (2021): 1352-1362.

Reviewer #3 (Remarks to the Author):

In this manuscript, Xia et al present SLAT, a method for the alignment of multiple slices of spatial transcriptomic (ST) data, designed to work cross-modality or -technology. By performing a cross-

dataset SVD combined with an adversarial learning strategy, SLAT learns a mapping between cells in separate slices. The authors demonstrate that the proposed method appears to outperform alternative methods for this task in speed and accuracy.

Overall, the content of the manuscript is quite interesting. The task of alignment of ST data has been recently emerging and there are currently few methods, and SLAT appears to be a meaningful improvement. However, there remains room to improve the degree to which the results, in terms of testing the method on a greater number of datasets and with significantly improved quantitative analysis. With these improvements, this manuscript will be a valuable addition to the literature.

Major points:

1. Overall, the assessment of the alignment is mostly qualitative based on visualizations of lines connecting cells. Comparisons of matching of cell types or regions does not seem like the most informative given that one could just write an algorithm to always match cells to others of the same type and region. For the benchmarking shown in Figure 2, instead of just a select individual example, there should be reported aggregated statistics over all pairs of slices (or a large number of pairings, etc.).
2. Quantitative comparison should also include an analysis of how well the alignment preserves neighborhoods – i.e., if two cells are very nearby in one slice how often are they aligned to cells that are nearby in the other slice? And this should also be compared across methods. I imagine this is the sort of metric in which a method like PASTE might perform better, but it is important to consider that such spatial consistency in the mapping is very desirable in an alignment.
3. Is it possible to take two copies of the same slice, but different rotated/perturbed, attempt to align them, and see what fraction of cells get aligned to their true equivalent in the other slice?
4. Similarly, the comparison shown in Fig. 3 should include reporting of quantitative analysis of the quality of the alignment, as well as a comparison to performance of other methods for this task.
5. Overall it kind of seems like none of the hyperparameters actually change anything when varied. Robustness is good but the investigation of hyperparameters should be varied into a regime where we see the model actually stops working well. For example, what is the point of the LGCN if 0 LGCN layers barely hurts the performance? It feels like there is either an issue with how the benchmark is done or the “joint accuracy” statistic is not actually very meaningful. (see major point 1).
6. It is possible to add more combinations of technologies for alignments, and study similarity and difference between different datasets. For example, using mouse brain datasets with MERFISH, slideseqV2, stereoseq, seq-FISH+?
7. Given that the Stereoseq embryo dataset covers 7 distinct time points, it would be highly desirable to be able to align across all time points. Is SLAT able to perform an alignment across all time slices, similar to that shown in SI Fig. 9? This would be very interesting to show/discuss in the main text.

Minor points:

1. Occasionally cells seem to get aligned to others on the complete opposite side of the domain (see for example Fig. 3b). Should this be interpreted as an error?
2. Line 57, abbreviation SVD should be defined at first use.
3. Fig. 4B is somewhat hard to read – the contrast of the ovals should be increased and the meaning of the roman numerals should be made more clear. Are they associated with the ovals?
4. Also for Fig. 4b shouldn't there be a quality score for both slices, not just one?

Detailed Responses to Reviewers' Comments

**Reviewer 1 Comments**

*The authors present a tool for aligning heterogeneous spatial slices using graph neural networks and*
*adversarial techniques. The paper showcases a substantial range of tested datasets and methods. However,*
*the evaluations and utility of the approach - absolutely central to any methods papers - are severely lacking.*
*This is not even just a major problem but substantially undercuts any contribution from the manuscript.*

Thanks for the constructive comments! We have improved and extended the evaluations extensively as
suggested (see below).

*1. The authors claim superior performance based on 'systematic benchmarks', but the results fall short of*
*supporting that claim. Apart from the runtime, the only evaluation metric provided is shown in Figure 2 and*
*Extended Data Figure 2. These figures present matrices that measure the alignment of region and cell type*
*annotations in two slices from the same dataset. However, these matrices are incorrectly labeled as confusion*
*matrices. To determine which method is superior, the reader must inspect the 300 lines connecting the two*
*slices and interpret the matrices. This approach leaves much to be desired in terms of clarity, rigor, and ease*
*of comparison.*

Thanks for the comments. The term “confusion matrix” was indeed inaccurate, which we have now renamed
to “contingency table” (new Fig. 2c). We chose to present alignment accuracy in the form of contingency
tables in an effort to emphasize the concept that spatial alignment should achieve accurate matching of both
the cell type and the spatial region. Meanwhile, we have now added more compact comparisons of the “joint
accuracy” metric in bar plots and line plots (new Extended Data Fig. 4b and 6), which is essentially defined as
the top right corner of the contingency table. Last but not least, “Cell type accuracy” and “Region accuracy”,
defined as vertical and horizontal marginalization of the contingency tables, are also presented in the form of
scatter plots (new Extended Data Fig. 4a). We hope that such revision could make the presentation clearer.

To improve the visualization of spatial alignments, we have also replaced the original “3D slice stacking”
plots with a new form of 2D spatial plot (new Fig. 2b), where each cell is colored by its alignment correctness
in both cell type and spatial region, which we believe is much more interpretable.

*2. Moreover, the remaining evaluations in the manuscript are even weaker as they lack any specific metrics*
*and rely solely on visually judging the 300 lines connecting the two aligned images. I wonder why the authors*
*did not test how well the alignment corrects the artificial rotation they applied for some of the comparisons.*

*Additionally, metrics such as micro or macro F1 scores or similar for region and cell-type matches may be*
*useful metrics.*

Thanks for the great suggestions. As detailed in response to comment #1, we have replaced the previous 3D
alignment visualization with a more interpretable 2D version (new Fig. 2b). In addition, we have tested how
well the alignment corrects the artificial rotation as suggested by the reviewer. Specifically, based on the
spatial alignment of each benchmarked method, we estimated the optimal corrective rotation angle by solving
the Procrustes problem¹, which is then compared with the ground truth angle. We found that all benchmarked
methods were reasonably accurate in this task, producing errors within just 3 degrees, except for STAGATE
which is slightly more deflected (new Extended Data Fig. 5). In particular, spatially unaware methods
(Harmony and Seurat) were equally accurate in the Visium and Stereo-seq slices but less accurate in the
MERFISH slices than spatially aware methods (SLAT and PASTE), which could be attributed to the fact that
the cell types and spatial regions are highly correlated in the Visium and Stereo-seq slices but not the
MERFISH slices^a.

It is worth noting that rotation correction is only suitable for evaluating the accuracy for homogeneous
alignments, but not heterogeneous alignments which may involve complex non-rigid deformations such as
those across embryogenesis timepoints (Fig. 4). As such, we retain the use of contingency tables and joint
accuracy as the primary evaluation metric.

We also added macro and micro F1 scores of cell type and region matching accuracy as suggested by the
reviewer (new Supplementary Fig. 4). SLAT matches both cell type and spatial region well (new
Supplementary Fig. 4c and 4d), and consistently outperforms in terms of joint accuracy (new Extended Data
Fig. 4b), joint macro F1, as well as joint micro F1 (new Supplementary Fig. 4a and 4b).

*3. Another important aspect to consider is that the evaluations primarily focus on aligning slices within a*
*homogeneous setting, meaning they are from the same dataset. Given the authors' claim of heterogeneous*
*alignment, I had anticipated more alignment across different datasets rather than just different slices or*
*timepoints. Nevertheless, it is commendable that they did align mouse fetal RNA-seq data with ATAC-seq*
*slices, albeit with the need for careful slice selection and manual cropping prior to applying their method.*
*However, once again, we lack clear metrics beyond the visual inspection of the figures depicting connecting*

^a. which renders spatial rotation estimable with just the correct cell type matching for Visium and Stereo-seq slices, but not MERFISH slides.

*lines to assess the quality of alignment. While the cross-modality ability is interesting, it does not appear to be*
*a widely applicable use case that would capture broad readership interest.*

Thanks for the comment. We'd first like to clarify that we define homogeneous alignment as the alignment of
morphologically conserved samples^b profiled with the same spatial omics technology, which can be well-
aligned by existing methods like PASTE². Apart from such, all other types of spatial alignments are defined as
heterogeneous alignment. This includes alignments that (1) span different technologies, with disparate spatial
resolution and technical confounding, (2) span multiple developmental timepoints, with non-rigid
deformations in spatial pattern, and (3) different molecular modalities. In particular, heterogeneous alignment
in the manuscript is not limited to that between Stereo-seq and spatial-ATAC-seq, but also includes seqFISH+
vs. Stereo-seq (Fig. 3a-c), and Visium vs. Xenium (new Extended Data Fig. 8), with the joint accuracy metric
calculated whenever applicable (new Supplementary Fig. 7).

With all due respect, we do not agree with the reviewer's view that cross-modality spatial alignment lacks
broad readership interest. Advances in spatial omics technologies have enabled profiling more and more
modalities with spatial information³⁻⁵, the alignment of which would certainly contribute to unraveling gene
regulation in the tissue context, analogous to what has been established in non-spatial technologies^{6,7}. We are
glad to present SLAT as the first futureproof spatial alignment method compatible with this setting.

*4. Furthermore, a significant point of concern is the existence of free parameters, such as the SVD*
*dimensionality. The dataset-specific values used for these parameters should be provided for each alignment.*
*It is possible that the lower performance of other methods could be attributed to more extensive fine-tuning of*
*the authors' methods compared to the default settings that were used for the other approaches. Extended*
*Figure 2 does address this to some degree with a metric that is not used in the main text, but it seems to have*
*a duplicated subpanel, which is concerning.*

Thanks for the reminder. In the benchmarks where aligned slices are homogeneous, SLAT use the default
setting for all hyperparameters: SVD dimensionality: 30, graph neighbor: 50, LGCN layer: 1, MLP hidden
layer dimension: 256, embedding hidden size: 2048, learning rate: 1e-4, dynamic clipping ratio: 0.6 (line 670
page 40).

In case studies where heterogeneous slices with different spatial resolutions are aligned, we changed the graph
neighbor size to adjust for the resolution difference and ensure comparable physical neighborhood across
slices, which are now clearly stated in the manuscript. Specifically, in the Visium vs. Xenium alignment, we

^b Typically, consecutive slices from the same tissue sample.

used $K = 5$ for the Visium slice and $K = 130$ for the Xenium slice (line 761 page 44); in the Stereo-seq vs.
seqFISH+ alignment, we used $K = 20$ for the Stereo-seq slice and $K = 50$ for the seqFISH+ slice (line 778
page 44); in the Stereo-seq vs. spatial-ATAC-seq alignment, we used $K = 50$ for the Stereo-seq slice and
$K = 20$ for the spatial-ATAC-seq slice (line 803 page 45); in the spatial-temporal mouse embryogenesis
alignment, we used the default $K = 50$ for both slices as the spatial resolutions are identical (line 816 page
46).

Meanwhile, to explore the possibility that fine-tuning the compared methods would also improve their
performance, we conducted hyperparameter search for all benchmark methods (new Supplementary Fig. 16).
The results suggest that these methods perform best with their default settings, except that STAGATE showed
slightly better performance when trained for 500 epochs (instead of the default 1000 epochs). After updating
the benchmark with these optimized hyperparameters, SLAT still achieved the best performance (new Fig. 2c
and new Extended Data Fig. 4).

The “joint accuracy” metric used in the original Extended Data Fig. 2 (now Extended Data Fig. 6) is the
proportion of cells with both cell type and spatial region matched correctly, which is essentially the upper
right corner of the contingency table. The metric is now formally defined in the manuscript (original line 631,
now line 708). We thank the reviewer for pointing out the panel duplication mistake, which has been fixed in
the updated manuscript.

*5. Additionally, Figure 2 fails to include testing of the Seurat and Harmony methods, which should have been*
*compared. Moreover, the authors should have considered downsampling datasets to ensure that all*
*algorithms could be evaluated, rather than excluding them due to memory limitations. This is particularly*
*crucial because PASTE (or the newer PASTE2) is the most similar method, yet the authors did not test it in*
*some cases.*

Thanks for the suggestions. The results of Seurat and Harmony were placed in the original Supplementary
Fig. 4 previously due to space limit, we have now move it to the main text (new Fig. 2) for ease of
comparison.

To evaluate the performance of PASTE in the largest Stereo-seq dataset, we subsampled it to varying sizes
from 200 to 102,400, and found that PASTE managed to complete the alignment when the cell number falls
below 25,600, but SLAT outperforms it by a large margin in all subsampled datasets (new Supplementary Fig.
5).

As an update to the original PASTE algorithm, PASTE2 mainly adds the support for partially overlapping
slices⁸. However, after running it in our benchmarks, we noticed that the update caused drastic performance
decline (Additional Figure 1). Considering that PASTE2 is still under development and not formally published
yet, we’d hesitate to report the performance evaluation in the manuscript.

**Additional Figure 1 Benchmarking result with PASTE2.** $n = 8$ repeats with different random seeds. Error bars
 indicate mean \pm s.d.

*Minor points:*

*Cell type is not usually considered a 'molecular feature', "aligning cells that share molecular features (e.g.,*
 *cell type)"*

Thanks for the reminder. We have corrected it in the revised manuscript (line 35 page 3).

**Reviewer 2 Comments**

*Xia et al. proposes SLAT, a computational method for aligning spatial omics datasets. SLAT begins with a*
 *batch correction preprocessing step via matrix decomposition. It then uses LGCN and MLP to learn an*
 *embedding. The embedding is encouraged to match across the two datasets by the design of a Wasserstein*
 *discriminator. After training, the embedding is used in combination with spatial information for matching.*
 *SLAT is benchmarked with other available methods including PASTE, STAGATE, Seurat and Harmony. It is*
 *implemented and tested across various datasets, showing its application in integration across platforms and*
 *modalities, inference of regulatory relationship and developmental trajectory. There are some questions*
 *regarding the method design, benchmark, and concepts.*

Thanks for the summary and encouraging comments!

*1. The batch removal step and the Wasserstein discriminator seem redundant as they both aim to integrate the*
 *embedding across conditions. Given the batch effect is removed by the preprocessing and SLAT uses a shared*
 *MLP layer, the necessity of the Wasserstein discriminator needs to be evaluated and better positioned,*
 *possibly through an ablation study.*

Thanks for the reminder. We'd first like to clarify that the batch removal step and the Wasserstein
discriminator serve different purposes by design, i.e., they correct for differences in the molecular and spatial
domain, respectively. The batch removal step corrects for technical confounding in the measured molecular
features without considering spatial context. Meanwhile, topological variations in the spatial graphs caused by
varying spatial properties such as spatial resolution would again deflect batch-corrected cell embeddings after
they pass through the LGCNs, necessitating a follow-up Wasserstein discriminator.

As suggested, we empirically verified the necessity of these components through ablation tests. While the
Wasserstein discriminator did not significantly improve the accuracy of SLAT in homogeneous alignments
where difference in the spatial domain is negligible (new Supplementary Table 2), it did boost accuracy
substantially in the heterogeneous alignment of Stereo-seq (0.22 μm) vs. spatial-ATAC-seq (20 μm) where the
spatial resolutions are drastically different (new Supplementary Table 3).

*2. In benchmarking, SLAT has better accuracy in identifying cells that are correctly matched by both region
and cell types. However, it seems that spatial-unaware methods (Seurat and Harmony) have better accuracy
in cell type matching alone compared with SLAT in most settings (Supplementary Fig 2, 4). Why counting the
cells that are correctly matched in both aspects serves as a better measure of matching than correct cell type
matching alone? From my understanding, correct cell type matching is already enough for a number of
certain tasks such as cell type label transfer.*

Thanks for the comment. While cell type matching is sufficient for cell type label transfer, such label transfer
is not the whole story. One example is multimodal regulatory inference based on Stereo-seq vs. spatial-ATAC-
seq alignment (new Extended Data Fig. 9c), where cells of different modalities are paired into multi-modal
metacells for integrative analysis. The metacells are typically defined at a much finer scale than cell types, and
can benefit from spatial information in their construction. While the spatially unaware method GLUE could
match cell types equally well between slices, SLAT matching is more accurate than GLUE in the spatial
domain. As a result, regulatory inference based on the spatially informed SLAT metacells allowed
identification of a key TF *Jund*⁹ in embryo heart development (new Extended Data Fig. 9c), which was missed
by GLUE (new Supplementary Fig. 9).

*3. There are other methods for matching across samples. For example, Mixscape [1] is a very simple method
that is used to match cells in scRNA samples by its nearest neighbors in the PCA space. Mixscape can serve
as a baseline for using spatial-unaware embedding to substitute the embedding learned by SLAT optimization.
This can also serve as an ablation study to show how the LGCN architecture helps for accurate matching.*

Thanks for the suggestion. We carefully inspected both the paper¹⁰ and the tutorial
(https://satijalab.org/seurat/articles/mixscape_vignette.html) of Mixscape, but found that Mixscape is a
method for estimating the perturbation status of single cells in Perturb-seq data through a Gaussian mixture

model, rather than a method that matches cells by their nearest neighbors in the PCA space, thus ineligible for
an alignment task.

Meanwhile, we have already reported the direct comparison between SLAT and established spatially unaware
algorithms (Seurat and Harmony, see new Fig. 2b and 2c, Extended Data Fig. 4), which can also be seen as an
ablation of spatial information. These experiments showed that spatially unaware algorithms tend to mismatch
spatial regions albeit matching cell types reasonably well, further confirming the necessity of spatial
information in spatial alignments.

*4. One-to-one or one-to-many. Is it really reasonable to establish a one-to-one matching while the measured*
*data is noisy and especially not of the same resolution? For example, in Mixscape [1], it is suggested to use*
*the mean of nearest neighbors to represent the counterfactual state. Tangram [2], a method for aligning*
*scRNA-seq and pixel-resolution spatial omics data, also considers a probabilistic matching scheme. In*
*general one-to-many matching may be a more reasonable task to denoise the matching itself and meaningfully*
*generalize to matching across technologies with different resolutions. Although this is mentioned in the*
*discussion part, the issue still needs to be better addressed due to its fundamental importance to the*
*methodology.*

Thanks for the insightful reminder! We agree with the reviewer that it is more meaningful to model spatial
alignment as probabilistic one-to-many matching, rather than exact one-to-one matching which does not exist
between different slices, especially heterogeneous ones with different spatial resolutions. Indeed, SLAT is able
to quantify the matching confidence of cells based on cosine similarity in the cell embedding space. So, we
are able to extend SLAT to one-to-many matching naturally. Based on an empirical NULL distribution
estimated by sampling 1,000 cell pairs from two slices randomly, matches with p -values less than 0.05 are
selected as the one-to-many probabilistic matching set (detailed in line 625 page 38 in the modified
manuscript).

We compared the performance of this one-to-many matching with the plain one-to-one setting by comparing
the accuracy of voting-based label transfer, and found that one-to-many matching improved joint accuracy in
the Visium and Stereo-seq datasets but undermined accuracy in MERFISH (Additional Figure 2). In
particular, one-to-many matching decreased the accuracy of region matching in MERFISH but not cell type
matching (Additional Figure 3). We speculate that this is because boundaries of the annotated anatomical
regions in the MERFISH slices are not clear-cut, as reflected by the relatively common region mismatches
(new Fig. 2b). One-to-many matching further smooths out the weak boundary signal, producing more region
mismatches.

For the fairness of comparison, we also expanded one-to-many matching to other methods (Additional Figure
2), and found that it consistently improved accuracy for STAGATE, while producing varying effect for Seurat

and Harmony. PASTE was not influenced by this change because its optimal transport framework always
 converges to a near singular one-to-one match. None of the other methods surpassed SLAT in accuracy
 regardless of matching mode.

**Additional Figure 2 Comparison of joint accuracy between one-to-one and one-to-many matching.** $n = 8$ repeats
 with different random seeds. Error bars indicate mean \pm s.d.

**Additional Figure 3 Comparison of cell type and region accuracy in the MERFISH dataset between one-to-one**
 **and one-to-many matching of SLAT.** $n = 8$ repeats with different random seeds. Error bars indicate mean \pm s.d.

One-to-many matching could be especially helpful when spatial resolution differs across slices. We exemplify
 this with the alignment between Visium and Xenium slices (original Extended Data Fig. 5, now Extended
 Data Fig. 8), where the Xenium slice contains over 100,000 cells with sub-cellular ($< 1\mu\text{m}$) resolution and the
 Visium slice contains only about 3,000 spots in $50\mu\text{m}$ resolution. In the SLAT one-to-many alignment, 6
 triple-positive Visium spots matched with more than 100 triple-positive cells in the Xenium slice, which is
 highly consistent with expert annotation. All other methods failed in comparison (original Extended Data Fig.
 5, now Extended Data Fig. 8).

*Minor points:*

*1. Typo in line 700. Harmony is mentioned twice. Should be STAGATE instead.*

Thanks for the kind reminder. We have corrected this mistake (line 771 page 44 of the modified manuscript).

*2. Supplementary Figure 4 should be better highlighted in the main text.*

Thanks for the kind reminder. We have merged the original Supplementary Fig. 4 to new Fig. 2.

*3. The rationale for using the matrix decomposition based batch correction instead of other batch correction*
*methods needs to be better elaborated.*

Thanks for the comment. Matrix decomposition is a common choice for batch correction¹¹. The SVD-based
method in SLAT is actually very similar to the first step of Seurat CCA, the only difference being that our
implementation keeps the eigenvalues while Seurat CCA does not. We favor this simple method over more
complex alternatives because of two reasons. First, SVD is fast and scalable. When aligning large slices
containing 100k cells, the combination of SVD and the SLAT core model costs under 3 minutes, while
Harmony itself takes ~6 minutes, and Seurat v3 takes over 10 hours (Fig. 2d). Using more complex batch
correction methods would drastically slow down SLAT, which is undesirable considering that the throughput
of spatial experiments is growing rapidly¹². Meanwhile, the SVD-based correction performs well with
comparable (and somewhat superior) joint accuracy over state-of-arts Harmony^{13,14} when being evaluated
during the development (new Extended Data Fig. 6c). In brief, we believe that the current SVD-based
correction procedure fits well with the overall design in terms of both speed and accuracy. We have
incorporated these into the Methods section of this revision (line 506 page 33).

*4. The evaluation metric used is only a minimum requirement for correct matching*

Thanks for the comment. We have incorporated additional evaluation metrics including rotation correction,
micro and macro F1 scores, and the edge score (see previous response to comment #2 of reviewer #1, and
comment #2 of reviewer #3 for more details).

*5. When performing benchmarking, the other methods are tested with their default parameters. In cases where*
*the performance is close, will fine-tuning other available methods yield a better result than SLAT?*

Thanks for the suggestion. We have now conducted hyperparameter search for all benchmarked methods as
we did for SLAT (new Supplementary Fig. 16, in response to comment #4 of reviewer #1). Most of the
methods work best under their default settings, except that STAGATE showed slightly better performance
when trained for 500 epochs rather than the default 1000 epochs. After re-running the benchmark with
updated hyperparameters, SLAT remains the top performer (new Fig. 2).

**Reviewer 3 Comments**

*In this manuscript, Xia et al present SLAT, a method for the alignment of multiple slices of spatial*
*transcriptomic (ST) data, designed to work cross-modality or -technology. By performing a cross-dataset SVD*
*combined with an adversarial learning strategy, SLAT learns a mapping between cells in separate slices. The*
*authors demonstrate that the proposed method appears to outperform alternative methods for this task in*
*speed and accuracy.*

*Overall, the content of the manuscript is quite interesting. The task of alignment of ST data has been recently*
*emerging and there are currently few methods, and SLAT appears to be a meaningful improvement. However,*
*there remains room to improve the degree to which the results, in terms of testing the method on a greater*
*number of datasets and with significantly improved quantitative analysis. With these improvements, this*
*manuscript will be a valuable addition to the literature.*

**Thanks for the encouraging comments!**

*1. Overall, the assessment of the alignment is mostly qualitative based on visualizations of lines connecting*
*cells. Comparisons of matching of cell types or regions does not seem like the most informative given that one*
*could just write an algorithm to always match cells to others of the same type and region. For the*
*benchmarking shown in Figure 2, instead of just a select individual example, there should be reported*
*aggregated statistics over all pairs of slices (or a large number of pairings, etc.).*

**Thanks for the suggestion. First, we'd like to mention that the previous 3D visualization of stacked slices has**
**now been replaced with 2D slice plots colored by matching correctness, hopefully improving its readability**
**(detailed in response to comment #1 of reviewer #1). While the visualization remains qualitative, we also**
**provide contingency tables and the joint accuracy metric to evaluate the alignments quantitatively.**

**Second, in real-world data analysis, cell type and spatial region annotations are often unavailable in one or**
**both of the slices being aligned. In the case of label transfer, annotating the slices is itself the purpose of slice**
**alignment. As such, the alignment method must not rely on the availability of cell type and region annotations.**
**An algorithm that simply matches annotated cell types and regions would not be very helpful in practice.**
**Meanwhile, we acknowledge that matching cells by cell types and region is not the finest criteria, but this is**
**an inevitable compromise given that the “ground truth alignment” of cells from different slices does not exist.**

**Third, we agree that evaluating the performance on all slices would better reflect the robustness of the**
**methods. Following the suggestion, we expanded our benchmark to all slices in the datasets we used, which**
**now encompass 12 Visium slices, 12 MERFISH slices and 4 Stereo-seq slices (new Extended Data Fig. 4).**

The results suggest that SLAT remains the best performing method in aggregated accuracy across multiple
slices.

*2. Quantitative comparison should also include an analysis of how well the alignment preserves*
*neighborhoods; i.e., if two cells are very nearby in one slice how often are they aligned to cells that are*
*nearby in the other slice? And this should also be compared across methods. I imagine this is the sort of*
*metric in which a method like PASTE might perform better, but it is important to consider that such spatial*
*consistency in the mapping is very desirable in an alignment.*

Thanks for the suggestion. The suggested metric is very similar to the “edge score” proposed by Joel et al¹⁵,
which is a metric originally used to evaluate the graph alignment. Specifically, the “edge score” is defined as
(also see Eqs. 19 in the modified manuscript):

$$A_{nm} = \begin{cases} 1, & \text{if } n' \sim m' \\ -1, & \text{if } n' \not\sim m' \end{cases}$$
$$\text{edge score} = \frac{1}{N} \sum_{n=1}^N \sum_{m \in \mathcal{N}_{(n)}} A_{nm}$$

where N is the number of cells in one slice, n and m are cell indices in this slice, $\mathcal{N}_{(n)}$ denotes the set of
neighbors of cell n , n' and m' are cells in the other slice aligned with cell n and m , respectively. “ \sim ”
means that the two cells are connected, i.e., they are spatial neighbors, while “ $\not\sim$ ” indicates the opposite.
Higher edge scores indicate that neighboring cells in one slice are more frequently matched with neighboring
cells in the other slice.

We calculated the edge score in all benchmark datasets (new Supplementary Fig. 14a), and found that PASTE
slightly outperforms SLAT. This is not unexpected because the Gromov-Wasserstein optimal transport
framework employed by PASTE is closely related to the idea behind the edge score, i.e., to preserve spatial
relation in one slice after mapping to the other slice. Nevertheless, high edge score does not always translate
to accurate alignment. For example, we found that despite PASTE getting the highest edge score in the
MERFISH dataset, only 29% of cells were aligned with correct cell types (new Fig. 2c). In comparison, SLAT
correctly aligned the cell type of 77% of the cells.

This discrepancy reveals an intrinsic limitation of the “edge score”, i.e., it is only a consistency metric, but
does not consider whether the alignment is biologically correct. For example, Supplementary Fig. 14b shows
two alignments with the same edge score, despite the fact that “Alignment 1” is all correct and “Alignment 2”
has four mismatches. So, we believe that such metric is not a reliable estimate of alignment quality, and favor
our label-based joint accuracy metric.

*3. Is it possible to take two copies of the same slice, but different rotated/perturbed, attempt to align them, and*
*see what fraction of cells get aligned to their true equivalent in the other slice?*

Thanks for the suggestion. We conducted the evaluation as suggested, while also noting that it is necessary to
add noise to the gene expression profile in the duplicated slice. Otherwise, the true match would be trivial to
find because it's the only cell that has the exact same gene expression values. Specifically, we added noise to
the duplicated slice by sampling from a negative binomial distribution centered at the measured expression
count. We set the inverse dispersion parameter to different values to simulate varying noise levels.

$$\text{NB}(x; \mu, \theta) = \frac{\Gamma(x + \theta)}{\Gamma(x)\Gamma(\theta)} \left(\frac{\mu}{\theta + \mu}\right)^x \left(\frac{\theta}{\theta + \mu}\right)^\theta$$

where μ and θ are the mean and inverse dispersion, respectively (also see Eqs. 17 in the modified
manuscript).

With increasing noise levels (from right to left along the x axis), SLAT and PASTE maintained high ground
truth accuracy, while the accuracy of Seurat and Harmony declined substantially (new Extended Data Fig. 2b).
STAGATE produced low accuracy across all noise levels. We noticed that PASTE always produces the perfect
matching even at very high noise levels, which is possibly due to the fact that the duplicated slices share the
exact same shape, which forms an unmistakable optimum in Gromov-Wasserstein transport of spatial
coordinates.

It is worth noting that this duplicate slice evaluation does not recapitulate the complexities in real-world
spatial alignments, where different slices would certainly have more or less deformation, rendering alignment
methods less accurate.

*4. Similarly, the comparison shown in Fig. 3 should include reporting of quantitative analysis of the quality of*
*the alignment, as well as a comparison to performance of other methods for this task.*

Thanks for the reminder. We have now added the quantitative evaluation of alignment quality in Fig. 3 using
the “joint accuracy” metric. The results again confirmed the superior accuracy of SLAT (new Supplementary
Fig. 7). Interestingly, we found that other spatially aware methods performed even worse than spatially
unaware methods in these heterogeneous alignments, possibly because they were designed for homogeneous
situations only and cannot handle the difference in spatial resolution/topology between heterogeneous slices.

*5. Overall it kind of seems like none of the hyperparameters actually change anything when varied.*
*Robustness is good but the investigation of hyperparameters should be varied into a regime where we see the*
*model actually stops working well. For example, what is the point of the LGCN if 0 LGCN layers barely hurts*
*the performance? It feels like there is either an issue with how the benchmark is done or the joint accuracy;*
*statistic is not actually very meaningful. (see major point 1).*

Thanks for the reminder. Setting LGCN layer to 0 essentially makes SLAT degenerate into a spatially unaware
algorithm. In the Stereo-seq dataset we used for the hyperparameter test, the cell type and spatial region are
highly correlated (new Supplementary Fig. 2), where spatially unaware methods could also achieve a
reasonable joint accuracy just by matching cell types correctly (new Extended Data Fig. 4), which explains
why setting LGCN layer = 0 did not affect performance.

Upon rerunning the hyperparameter test of LGCN layers on the MERFISH dataset, where cell type and spatial
region are less correlated (new Supplementary Fig. 2), we found substantial difference between LGCN layer =
0 and LGCN layer = 1 (new Extended Data Fig. 6a), especially in terms of region accuracy (new
Supplementary Fig. 18), which confirms the necessity of LGCN in making use of spatial information.

In the meantime, we also expanded the range of key hyperparameters to identify their working interval (new
Extended Data Fig. 6a).

*6. It is possible to add more combinations of technologies for alignments, and study similarity and difference*
*between different datasets. For example, using mouse brain datasets with MERFISH, slideseqV2, stereoseq,*
*seq-FISH+?*

Thanks for the suggestion. While we are indeed very interested in the mouse brain and wish to incorporate
more diverse technologies, we found that different spatial omics studies sampled at very different brain
regions. For example, Zhang et al. studied the **primary motor cortex** using MERFISH¹⁶, Shah et al. studied
the **temporal and parietal cortex** using seqFISH¹⁷, Takei et al. studied the **cerebral cortex** using
seqFISH+¹⁸, VIZGEN profiled the **coronal** slices of **whole brain** using MERFISH, while Chen et al. profiled
**sagittal** slices using Stereo-seq¹⁹. As for Slide-seqV2, its only application in the mouse brain is at the
**hippocampus**²⁰. As these slices come in different brain regions and orientations, they cannot be meaningfully
aligned.

With the accumulation of more spatial omics data as well as the emergence of standardized sampling
protocols such as the common coordinate framework (CCF)²¹, SLAT would greatly facilitate spatial alignment
in brain data and beyond. We plan to continue to update our demos/tutorials at
<https://slat.readthedocs.io/en/latest/tutorials.html> as more suitable data become available.

*7. Given that the Stereo-seq embryo dataset covers 7 distinct time points, it would be highly desirable to be*
*able to align across all time points. Is SLAT able to perform an alignment across all time slices, similar to*
*that shown in SI Fig. 9? This would be very interesting to show/discuss in the main text.*

Thanks for the nice suggestion. We additionally applied SLAT to align mouse embryo slices spanning the
developmental stages of E9.5–E16.5, encompassing a total of 8 timepoints (new Supplementary Fig. 11a and
11b).

For the ease of inspection, we use Sankey plot to present the cell type development across all timepoints as
inferred from the alignment (new Supplementary Fig. 11c). The SLAT alignment accurately recapitulated the
development of different organs, consistent with existing knowledge about mouse embryo development and
the original authors' manual annotation (see Fig. S2D of original paper¹⁹). It is worth noting that SLAT took
only about 15 minutes to align the 462,454 cells across all 8 slices (using 32 cores and 1 GPU on a single
server), demonstrating its unique power in large-scale spatial alignments. These results have been
incorporated in the main text (line 241 page 13 and new Supplementary Fig. 11).

*Minor points:*

*1. Occasionally cells seem to get aligned to others on the complete opposite side of the domain (see for*
*example Fig. 3b). Should this be interpreted as an error.*

Thanks for the reminder. Close inspection showed that these cells are in fact “neural crest” cells which
migrates throughout the embryo (original Extend Data Fig. 3a, now Extend Data Fig. 7a). We have
incorporated this into the revised legend of Fig. 3b.

*2. Line 57, abbreviation SVD should be defined at first use.*

Thanks for the reminder. We have added the definition.

*3. Fig. 4B is somewhat hard to read; the contrast of the ovals should be increased and the meaning of the*
*roman numerals should be made clearer. Are they associated with the ovals?*

Thanks for the reminder. We have increased the contrast of the ovals. Each roman numeral corresponds to an
oval, and we have added arrows to clarify the correspondence between them (new Fig. 4b).

*4. Also for Fig. 4b shouldn't there be a quality score for both slices, not just one?*

Thanks for the reminder. There are indeed quality scores for both slices. We only showed one for illustration
purpose. Scores for the other slice is now shown in new Extended Data Fig. 10c.

Reference

1. Kabsch, W. A solution for the best rotation to relate two sets of vectors. *Acta Cryst A* **32**, 922–923 (1976).

2. Zeira, R., Land, M., Strzalkowski, A. & Raphael, B. J. Alignment and integration of spatial transcriptomics

data. *Nat Methods* **19**, 567–575 (2022).

3. Goltsev, Y. *et al.* Deep Profiling of Mouse Splenic Architecture with CODEX Multiplexed Imaging. *Cell*

174, 968–981.e15 (2018).

4. Deng, Y. *et al.* Spatial-CUT&Tag: Spatially Resolved Chromatin Modification Profiling at Cellular Level.

Science **375**, 681–686 (2022).

5. Zeng, H. *et al.* Spatially resolved single-cell translomics at molecular resolution. *Science* **380**, eadd3067

(2023).

6. Argelaguet, R., Cuomo, A. S. E., Stegle, O. & Marioni, J. C. Computational principles and challenges in

single-cell data integration. *Nat Biotechnol* **39**, 1202–1215 (2021).

7. Ma, A., McDermaid, A., Xu, J., Chang, Y. & Ma, Q. Integrative Methods and Practical Challenges for

Single-Cell Multi-omics. *Trends Biotechnol* **38**, 1007–1022 (2020).

8. Liu, X., Zeira, R. & Raphael, B. J. PASTE2: Partial Alignment of Multi-slice Spatially Resolved

Transcriptomics Data. 2023.01.08.523162 Preprint at <https://doi.org/10.1101/2023.01.08.523162> (2023).

9. Ricci, R. *et al.* Distinct functions of junD in cardiac hypertrophy and heart failure. *Genes Dev* **19**, 208–213

(2005).

- 10. Papalexi, E. *et al.* Characterizing the molecular regulation of inhibitory immune checkpoints with
multimodal single-cell screens. *Nat Genet* **53**, 322–331 (2021).
- 11. Liu, Y., Wang, T., Zhou, B. & Zheng, D. Robust integration of multiple single-cell RNA sequencing
datasets using a single reference space. *Nat Biotechnol* **39**, 877–884 (2021).
- 12. Moses, L. & Pachter, L. Museum of spatial transcriptomics. *Nat Methods* **19**, 534–546 (2022).
- 13. Korsunsky, I. *et al.* Fast, sensitive and accurate integration of single-cell data with Harmony. *Nature*
*Methods* **16**, 1289–1296 (2019).
- 14. Tran, H. T. N. *et al.* A benchmark of batch-effect correction methods for single-cell RNA sequencing
data. *Genome Biology* **21**, 12 (2020).
- 15. Douglas, J. *et al.* Metrics for Evaluating Network Alignment. 6.
- 16. Zhang, M. *et al.* Spatially resolved cell atlas of the mouse primary motor cortex by MERFISH. *Nature*
**598**, 137–143 (2021).
- 17. Shah, S., Lubeck, E., Zhou, W. & Cai, L. In situ transcription profiling of single cells reveals spatial
organization of cells in the mouse hippocampus. *Neuron* **92**, 342–357 (2016).
- 18. Takei, Y. *et al.* Single-cell nuclear architecture across cell types in the mouse brain. *Science* **374**, 586–594
(2021).
- 19. Chen, A. *et al.* Spatiotemporal transcriptomic atlas of mouse organogenesis using DNA nanoball-
patterned arrays. *Cell* **185**, 1777–1792.e21 (2022).

20. Stickels, R. R. *et al.* Highly sensitive spatial transcriptomics at near-cellular resolution with Slide-seqV2.

*Nat Biotechnol* **39**, 313–319 (2021).

21. Yao, Z. *et al.* A high-resolution transcriptomic and spatial atlas of cell types in the whole mouse brain.

2023.03.06.531121 Preprint at <https://doi.org/10.1101/2023.03.06.531121> (2023).

Reviewer #1 (Remarks to the Author):

I appreciate the authors' responses. Although the authors have introduced metrics for the homogeneous alignment case, the manuscript still lacks evaluations for heterogeneous applications. This is pretty important for the manuscript. I'd suggest that the authors should define and quantitatively validate performance in at least one case like this.

For the cell type evaluation, there is a large difference between macro and micro F1 scores, this suggests rare cell types are not being aligned correctly. This is particularly concerning as less common cell types are often of greater interest for scientists. To shed light on these differences, the inclusion of confusion matrices would be highly beneficial, allowing readers to examine the results for specific cell types.

Reviewer #4 (Remarks to the Author):

The authors have satisfactorily addressed the reviewers' comments and made necessary changes.

Reviewer #3 (Remarks to the Author):

In this revision the authors have added some additional results and visualizations and have addressed a majority of the points raised in the previous round. However, there remain some questions about the benchmarking, as well as some things that should be commented on in the discussion of the manuscript.

Specific follow-up comments to previous points:

1. The addition of statistics over multiple slices is appreciated and quite helpful for showing the performance of SLAT. However, at least some statistics over multiple slices, including statistical significance of the differences, should be presented in the main manuscript, instead of just the SI.
4. As mentioned in the rebuttal, the LGCN = 0 case should basically remove information, as should the noise \rightarrow 1 case (I believe). Since SLAT only seems to lose at most around 10 points of accuracy in these cases, is most of the alignment coming from the SVD element instead of the GCN? This should be more clearly addressed/discussed.
6. Could the authors at least discuss the potential applications of SLAT to align datasets from different technologies, if there are no suitable datasets for computational benchmark at this time?

Minor comments:

1. The color schemes in Fig. 2b and extended data Fig. 2 are very hard to follow/distinguish visually.
2. Extended data Fig. 3 – typo in "Merfish"

Detailed Responses to Reviewers' Comments

Reviewer 1 Comments

I appreciate the authors' responses.

Thanks for the encouraging feedback!

Although the authors have introduced metrics for the homogeneous alignment case, the manuscript still lacks evaluations for heterogeneous applications. This is pretty important for the manuscript. I'd suggest that the authors should define and quantitatively validate performance in at least one case like this.

Thanks for the nice suggestion! The major obstacle that hinders evaluation of heterogeneous alignments beyond regular discrete category-based metrics is the non-rigid deformation (e.g., Extended Data Fig. 7a, b for seqFISH and Stereo-seq mouse embryo slice alignment). Here, we devised an anatomical knowledge-based approach in attempt to sidestep this problem. Briefly, we first manually curated a set of 7 “key points” in the two slices corresponding to distinctive anatomical structures (jaw, eye, brain, spine, heart, back and tail) that should be aligned together. We then calculated the (Euclidean) distance between the aligned cell and the ground truth for each key point pair, taking the average across all key points as the overall “key point deviation distance” (Additional Figure 1a and b). We found that SLAT obtained lower key point deviation distance than others (Additional Figure 1c and d), suggesting a higher alignment accuracy.

Additional Figure 1 Illustration of the proposed key point-based evaluation for heterogeneous alignment in the seqFISH vs. Stereo-seq case. **a**, Illustration of the key point deviation distance. **b**, Curated key points in the seqFISH and Stereo-seq datasets. **c**, Visualization of the alignment produced by different algorithms. Red points are key points while blue points are cells aligned by the algorithms. Green solid lines indicate ground truth matching of key points and grey dashed lines indicate the algorithmic alignment. **d**, Average key point deviation distance of different methods.

For the cell type evaluation, there is a large difference between macro and micro F1 scores, this suggests rare cell types are not being aligned correctly. This is particularly concerning as less common cell types are often of greater interest for scientists. To shed light on these differences, the inclusion of confusion matrices would be highly beneficial, allowing readers to examine the results for specific cell types.

Thanks for the reminder on the notable gap between macro F1 and micro F1 (by ~ 0.15) for MERFISH and Stereo-seq dataset! As suggested, we added the cell-type confusion matrices (Additional Figure , which has been incorporated as the new Supplementary Fig. 5 in this revision) to ease the assessment of cell type-specific accuracies, which suggests, just as the referee pointed out correctly, a contribution of rare cell types to the gap between macro and micro F1.

Briefly, several rare cell types in the MERFISH dataset tend to be aligned to transcriptionally similar, but different, cell types by SLAT¹. Meanwhile, we also noticed that SLAT, but not PASTE and STAGATE, could effectively align rare cell types with more distinct expression profiles like “Pericytes” and “Microglia”, suggesting that SLAT’s combination of graph convolution and adversarial learning could integrate spatial context and molecular features more effectively.

On the other hand, we also noticed that, in the Stereo-seq dataset, the “mismatches” could mostly be attributed to inconsistent cell type taxonomies in the source annotation, e.g., cells annotated as various organs (including rare ones) were partly aligned with “connective tissue”, because there is indeed connective tissue in most organs, in which case “mismatches” are expected and do not indicate inaccuracy of the alignment methods.

We have incorporated these into the revised manuscript (Line 136, Page 8).

¹ For instance, the 5 “OD mature 3” cells (out of 6 in total) and 8 “OD mature 4” cells (out of 10 in total) were aligned with “OD mature 2” by SLAT.

SLAT

PASTE

STAGATE

Additional Figure 2 Confusion matrix of cell type matching for SLAT, PASTE and STAGATE (see Supplementary Fig. 5 for the complete results).

Reviewer 4 Comments

None

Reviewer 3 Comments

In this revision the authors have added some additional results and visualizations and have addressed a majority of the points raised in the previous round.

Thanks for the encouraging comments!

However, there remain some questions about the benchmarking, as well as some things that should be commented on in the discussion of the manuscript. The addition of statistics over multiple slices is appreciated and quite helpful for showing the performance of SLAT. However, at least some statistics over multiple slices, including statistical significance of the differences, should be presented in the main manuscript, instead of just the SI.

Thanks for the reminder! We have now added statistical test (paired Wilcoxon rank sum test) of the differences on Visium ($n=9$) and MERFISH ($n=11$). The updated results were moved to new Fig. 2d. Stereo-seq alignments were unsuitable for statistical tests because the number of slice pairs was too small ($n=2$).

As mentioned in the rebuttal, the LGCN = 0 case should basically remove information, as should the noise \rightarrow 1 case (I believe). Since SLAT only seems to lose at most around 10 points of accuracy in these cases, is most of the alignment coming from the SVD element instead of the GCN? This should be more clearly addressed/discussed.

Thanks for the comments! While the performance decrease upon LGCN ablation or spatial graph corruption is relatively mild in the Visium and Stereo-seq slices, the impact on MERFISH slices was more substantial, i.e., 0.171 points (38.5%) decrease in joint accuracy (Additional Table 1, Extended Data Fig. 6a, b), which in particular stems from the >20 points decrease in spatial region accuracy (Supplementary Fig. 19).

	10x Visium	MERFISH	Stereo-seq
Seurat	0.518	0.256	0.629
Harmony	0.492	0.256	0.637
SLAT (LGCN = 0)	0.638 (-12.1%)	0.273 (-38.5%)	0.633 (-9.2%)
SLAT	0.704	0.444	0.696

Additional Table 1 Joint accuracy of different methods in three benchmark datasets.

Briefly, these data suggested that the different impact of the LGCN layer is largely dependent on the “correspondence” between the spatial regions and cell types of input data: in the Visium and Stereo-seq slices where cells with the same/similar cell types occupy contiguous spatial regions (as is evident in the high spatial-neighbor cell type consistence, Supplementary Fig. 2), a spatially unaware aligner (e.g., Seurat, Harmony and our SVD component) could just match spatial regions fairly well based on the cell types only. However, for the MERFISH slices where regions and cell types are less predictive of each

other (Supplementary Fig. 2), the alignment is not as accurate without LGCN, which demonstrates the necessity of encoding spatial information in such alignments.

We have incorporated these into the revised manuscript (Line 758, Page 44).

Could the authors at least discuss the potential applications of SLAT to align datasets from different technologies, if there are no suitable datasets for computational benchmark at this time.

Thanks for the suggestion! We have added a discussion of potential applications in aligning different technologies in the Discussion section (Line 289, Page 14).

Reviewer #1 (Remarks to the Author):

I thank the authors for addressing my remaining remarks.